# Asset-level assessment of climate physical risk matters for adaptation finance

Giacomo Bressan[1], Anja Đuranović [2], Irene Monasterolo [1,2,3] &
Stefano Battiston [4,5]

Climate physical risk assessment is crucial to inform adaptation policies and finance. However, science-based and transparent solutions to assess climate physical risks are limited, compounding the adaptation gap. This is a main limitation to fill the adaptation gap. We provide a methodology that quantifies physical risks on geolocalized productive assets, considering their exposure to chronic and acute impacts (hurricanes) across the scenarios of the Inter-governmental Panel on Climate Change. Then, we translate asset-level shocks into economic and financial losses. We apply the methodology to Mexico, a country highly exposed to physical risks, recipient of adaptation finance and foreign investments. We show that investor losses are underestimated up to 70% when neglecting asset-level information, and up to 82% when neglecting tail acute risks. Therefore, neglecting the asset-level and acute dimensions of physical risks leads to large errors in the identification of adaptation policy responses, investments and finance tools aimed to build resilience to climate change.

The analysis of climate physical risks, meant as both acute risks stemming from climate-related hazards and chronic risks from long-run climate impacts[1], plays a key role to inform decision-makers[2] and build resilience to climate change[3]. However, methodological and conceptual challenges remain open for assessing the impact of climate physical risks[4], in particular on business and finance[5,6]. Addressing these challenges is crucial to identify policy responses[7], financing needs[8] and the financial instruments[9] to fill the adaptation investment gap. In particular, the underestimation of climate physical risks leads to underinvesting in adaptation and mitigation[10], which in turn leads to delayed climate action, larger socio-economic losses and higher risks. Indeed, while adaptation finance has increased from 30 billion (annual) USD to 46 billion in the period 2017–2020[8], it still falls short of needs, estimated to amount to 250 billion USD per year for low-income and emerging countries alone by 2030[11]. At the same time, the adaptation finance needs for other countries and regions, including the European Union's member states, have still to be clearly identified.

Most often, the assessment of climate physical risks relies on commercial methodologies. Nevertheless, these methodologies are proprietary (and thus not easily accessible), not fully transparent, not fully replicable, and often lead to diverging results[12]. Moreover, commercial methodologies provide climate risk scores with different levels of aggregation (e.g., by firm or hazard, depending on the provider). While scores could be useful for some types of analyses (e.g., investigating the market premium for physical risk), they cannot be used as a proxy for the asset-level damages, which in turn are needed to inform climate financial valuation models and climate financial risk pricing.

To address these limitations, here we develop a methodology for asset-level climate physical risk assessment that translates the impact of both acute and chronic risk on the financial value of firms with productive assets exposed to climate impacts, and on financial securities (equity) associated with those firms. We apply the methodology to a sample of 177 listed firms, owning 1820 physical assets located in Mexico, a country that is highly exposed to physical risks and is a main

[1]Institute for Ecological Economics, Vienna University of Economics and Business, Vienna, Austria. [2]Utrecht University School of Economics, Faculty of Law, Economics and Governance, Utrecht University, Utrecht, The Netherlands. [3]Centre for Economic Policy Research (CEPR), London, UK. [4]Department of Banking and Finance, University of Zurich, Zurich, Switzerland. [5]Research Institute for Complexity, Department of Economics, Ca' Foscari University of Venice, Venice, Italy. ✉e-mail: i.monasterolo1@uu.nl; stefano.battiston@df.uzh.ch

beneficiary of adaptation finance[3]. In the absence of granular data on industrial plants and their ownership, previous works on risk assessment have resorted to approximating the location of economic activities of each firm with its headquarter location. Still, the magnitude of the resulting misestimation of losses using such proxy data has been unknown so far. Here, we show that, in our sample, the approximation leads to an underestimation of climate-related losses of up to 70%, in terms of relative difference on Value at Risk (VaR), compared to the computation based on granular asset-level data. Further, current analyses of physical risks rely either on scenarios of chronic risks (e.g., sea-level rise or water scarcity), or, separately, of acute risks (e.g., cyclones[13] or floods). Focusing on either acute or chronic risks alone can lead to an underestimation of losses, but the magnitude of such underestimation is poorly understood[6]. Here, we show that neglecting the component of acute risk can lead to underestimations of financial losses for investors up to about 82% in terms of relative difference on VaR. By quantifying these potential underestimations, our methodology enables investors and corporations to better integrate climate physical risks in their internal risk assessment and risk management processes. Finally, the literature has mainly focused on the impact of past climate hazards on current prices of financial assets and risk premia[14–16], while the information on future climate scenarios is not integrated. In contrast, our model provides a scenario-contingent financial valuation of firms that accounts both for historical information (embedded in current prices) as well as future climate scenarios. In particular, we examine climate acute and chronic risk depending on the type and location of the firm's production plants.

## Results
### Methodological framework
We introduce a methodology to perform the assessment of climate physical risks at the asset level for individual firms and investors' portfolios. We define "assets" as the facilities such as mines, power plants, cement factories, etc., which are operated by a firm. "Asset-level data" represent a set of information collected on such plants, which include location, capacity, value, prices, residual life, etc. A firm has "available asset-level data" if information about plants that are relevant for its business is available. Data on production plants' type, geolocation, and capacity are often either missing or fragmented, despite recent progress in spatial finance (https://www.cgfi.ac.uk/spatial-finance-initiative/)[17]. Furthermore, plants' owners identity is not standardised, limiting researchers' ability to reconstruct the ownership chains from plants to firms and investors, due to the complexity of the global ownership network[18,19]. Here, we provide a procedure that leverages existing plant data and cross-matches owners' identity in the main existing datasets.

With respect to climate risks, we consider both the chronic and acute components individually and combined, conditional to the Intergovernmental Panel on Climate Change (IPCC) climate scenarios, identified as combinations of Representative Concentration Pathways (RCP) and Shared Socioeconomic Pathways (SSP). For chronic risk, we use point estimates of losses from a macroeconomic model (see "Methods"). For acute risk, we consider both the average loss and the tail of the loss distribution (extreme events). Following the terminology of the relevant literature, the average yearly loss is referred to as Expected Annual Impact (EAI) while tail risk is referred to in terms of return period. For instance, a 100-year return period (RP100) indicates a value of losses that on average is exceeded only every 100 years. Equivalently, RP100 is a loss that is exceeded with a probability of 1% (or, in statistical terms, a 0.99-quantile). Return periods are a measure of losses (not a measure of time) and are equivalent to quantiles of the loss distribution, under the condition that both are computed on the same sample. Consistently with the risk management literature, this notion of tail risk corresponds to the Value at Risk (VaR) at a given confidence level. For instance, the 0.99-quantile corresponds to a 99% VaR.

The methodology is articulated in five steps, which are represented in Fig. 1. The first block is the database model for asset-level and business lines data, ownership chain, and financial information (see "Methods" for a full description). The second block represents the probabilistic climate acute risk assessment at the asset level that is performed using the CLIMADA model[20,21] to assess hurricane damages. The third block connects the acute impacts with the sector-level chronic impacts on firms' business lines, which are computed with a macroeconomic model (i.e., the Intertemporal Computable Equilibrium System (ICES) model)[22–24]. The fourth block connects asset-level impacts with the performance of the firm owning the asset, and translates them into an adjustment of the financial valuation of the securities (equity) issued by the firm by developing a Climate Dividend Discount Model (CDDM), that builds on standard equity valuation theory[25–29]. The adjustment is calculated with respect to a baseline in which only present acute and chronic risks are accounted for. Finally, in the fifth block, climate physical risks are translated into financial risk metrics for the investor who holds the firm's securities.

### Acute risks at the asset level
We use CLIMADA to perform a probabilistic risk assessment of tropical cyclones' damages to physical assets in Mexico. Firms' assets are heterogeneously distributed in the country, as shown in Fig. 2a. Assets are also heterogeneous in terms of sector and productive capacity. In our sample, mines and power plants are the most represented assets, and are particularly concentrated on the western coast of the country and in the central region. Relatively few assets are present on the eastern coast, especially in the south. Direct damages from tropical cyclones lead to economic losses that are measured as a percentage of assets' values. Expected Annual Impacts (EAI) for year 2050 conditioned to scenario RCP2.6 are low in the whole country, as shown in Fig. 2b. As a first comparison, tail risk losses, i.e., RP100 (Fig. 2c), are at least ten times larger than EAI for 37% of assets (in number), conditioned to the same mild scenario RCP2.6. However, losses from tropical cyclones are concentrated in certain areas of the country, such as the Jalisco region, as indicated by the yellow, orange, and red colours in Fig. 2c, d. Indeed, other areas, such as the central region, are not exposed to tropical cyclones as indicated by the grey colour. Finally, we look at more extreme tail risk, i.e., a higher return period (RP250), in a more severe climate scenario (RCP4.5): Fig. 2d shows that, in this case, more numerous assets suffer larger losses. We find that for about 73% of assets (in number) RP250 losses are at least ten times larger than EAI. Moreover, for 13% of assets, RP250 losses are five times larger than RP100. Note that the ratio of RP250 losses RP100 losses cannot be computed for a subset of 631 assets in the sample with zero RP100 but non-zero RP250. Differences in losses across scenarios are limited due to the short time horizon considered, but become more pronounced toward the end of the century. Thus, it is crucial to take into account both the RCP scenario and higher return periods to properly quantify tail acute risks at the asset level.

### Acute vs chronic risks at the firm level
The equity value of the firm is a function of chronic and acute losses (which are modelled as random variables, see Methods). In line with the finance literature, this function is referred to as equity valuation. We first compute the equity value considering future chronic and acute risks (conditioned to climate scenarios). Then, we compare this first value with the result of the computation where only current levels of chronic and acute risks are accounted for, i.e., a value that only reflects the current distribution of losses on assets and business lines. Finally, the equity shock is computed as the relative difference between the two values. Hence, the equity shock represents the adjustment in equity valuation when considering the impact of climate scenarios on physical assets and business lines. In Fig. 3, we show the impact of acute and chronic shocks on the equity value in a sample of 86 firms with asset-

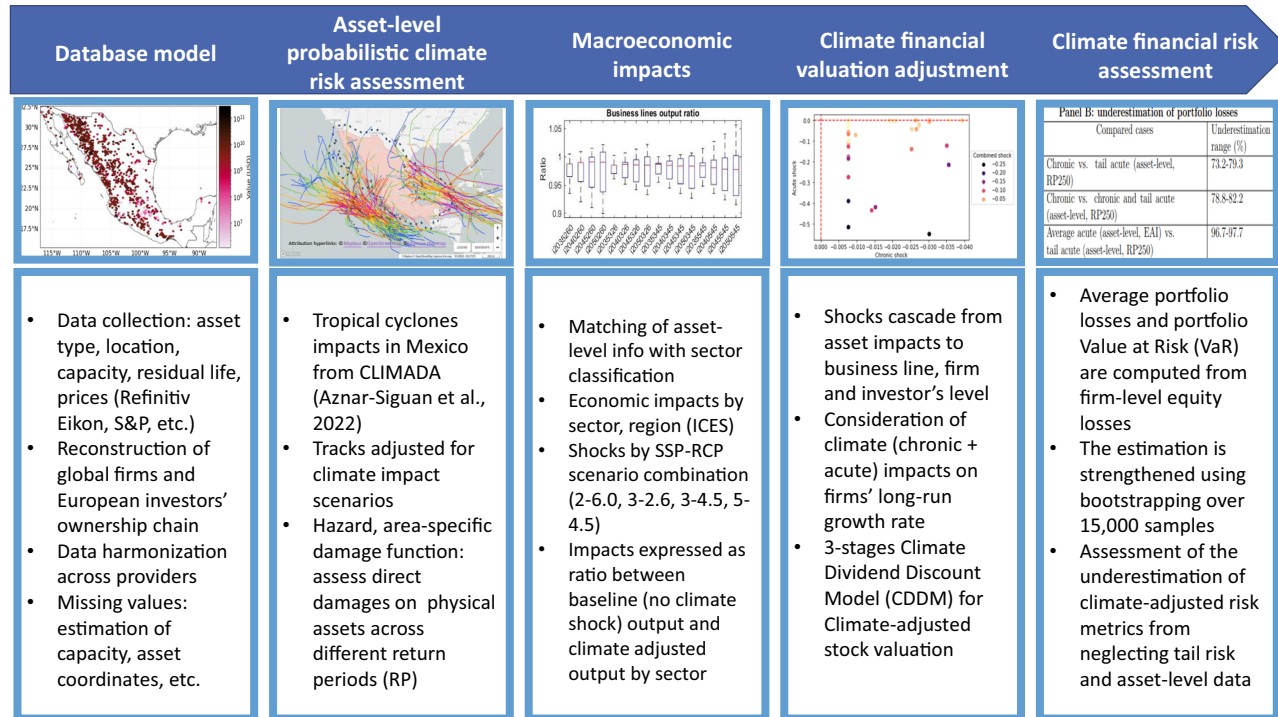

**Fig. 1 | Methodological framework of the asset-level approach to climate physical risk assessment and financial valuation.** From left to right, the blocks represent consequential steps of implementation. We start with asset-level data collection and harmonization, and then we feed an asset-level probabilistic risk assessment for tropical cyclones. Then, we compute the sector-based and macro-economic impacts of physical risk using the Intertemporal Computable Equilibrium System (ICES) macroeconomic model. By combining climate impacts at the asset level and the resulting macroeconomic impacts, it is possible to quantify the climate financial impacts at the firm level. Finally, we adjust the financial valuation for equity contracts of the firm that owns the plants, and compute the resulting adjustment in financial risk for the firm's investor. Source: authors. For the map in the second square from the left (below Asset-level probabilistic climate risk assessment): authors' elaboration on NOAA, historical hurricane tracks data (https://coast.noaa.gov/hurricanes), underlying map from Mapbox (https://www.mapbox.com/about/maps, https://labs.mapbox.com/contribute/), and Open-StreetMap (https://www.openstreetmap.org/about/).

level data, conditioned to climate scenario SSP3-RCP4.5. Acute shocks represent the RP250 losses on the assets of the firm while chronic shocks represent the point estimate of losses from the macroeconomic model. The colour code represents the 99.6% VaR of the equity shock. Four main results emerge. First, chronic and acute shocks both contribute to higher equity shocks (see e.g., firm 1, mainly active in the fossil fuels sector). Second, firms in the same sector can have similar chronic shocks but very different acute shocks because their assets are in different locations (see e.g., firm 2 vs firm 3, active in the renewables sector). Third, firms may be affected by a similar acute shock but by different chronic shocks (see e.g., firm 3 and 4, active in the renewables and mining sectors, respectively). Fourth, firms can have large acute shocks even if operating in different sectors (see e.g., again firms 3 and 4).

### Tail acute risks lead to large economic losses for firms
Figure 4 illustrates the relation between impacts on firms' equity in terms of tail risk (RP250) and average losses (EAI), with negative (positive) values representing losses (gains). As RP250 losses are always more negative or equal to EAI losses, dots in the scatter plot are below the black line, representing points where EAI losses equal RP250 losses.

Note that for 35% of firms in the sample RP250 is at least three times larger than EAI. This result is robust to changes in value of key parameters in the model such as discount rate $r$ and long-term dividend growth rate $g_L$ (see Supplementary Section 7).

### Substantial underestimation of investors' losses from neglecting tail acute risks
We now look at how firm-level losses translate into portfolio losses. To this aim, we consider an investor owning an equally weighted portfolio invested in the stocks issued by the firms in our sample. So

far, by means of the valuation model described in "Methods", for each firm we have computed firm-level equity losses, both in terms of EAI and RP250. We now compute portfolio losses for the investor as the equally weighted average of all firm-level equity losses. In particular, the equally weighted average of firm-level EAI and RP250 are interpreted as the average portfolio loss and the 99.6% VaR on the investor's portfolio, respectively. Table 1A shows the point estimates and the confidence intervals of portfolio losses for five different combinations of loss metric and type of physical risk. These include chronic risk only, acute risk only computed using EAI, acute risk only computed using RP250, the combination of chronic and acute risk computed using EAI, and the combination of chronic and acute risk computed using RP250. We compare cases that consider tail risk (RP250) with cases that do not consider tail risk (EAI). The confidence intervals are computed using the bias-corrected and accelerated percentile method (see e.g., ref. 30) on 15,000 samples. In the absence of precise rules to set the number of samples for the boot-strap, we determined it iteratively. We started from a small number of samples and computed the confidence intervals. Subsequently, we increased the number of samples until we reached a point where the additional increase in the precision of the estimates was negligible. The bootstrapped distributions of sample means are shown in Supplementary Fig. 6. We highlight the importance of tail risk by comparing the magnitude of the VaR with the magnitude of the mean in Table 1A. In fact, the VaR is between 5 and 38 times the mean (comparing rows 3 and 5 (VaR) with rows 2 and 4 (mean), respectively). In Table 1B, we quantify the underestimation of portfolio risk stemming from neglecting tail risk. Risk underestimation is substantial. In fact, using chronic risk only instead of tail acute risk leads to an underestimation of losses up to 82.2% (row 2, column 3). At the

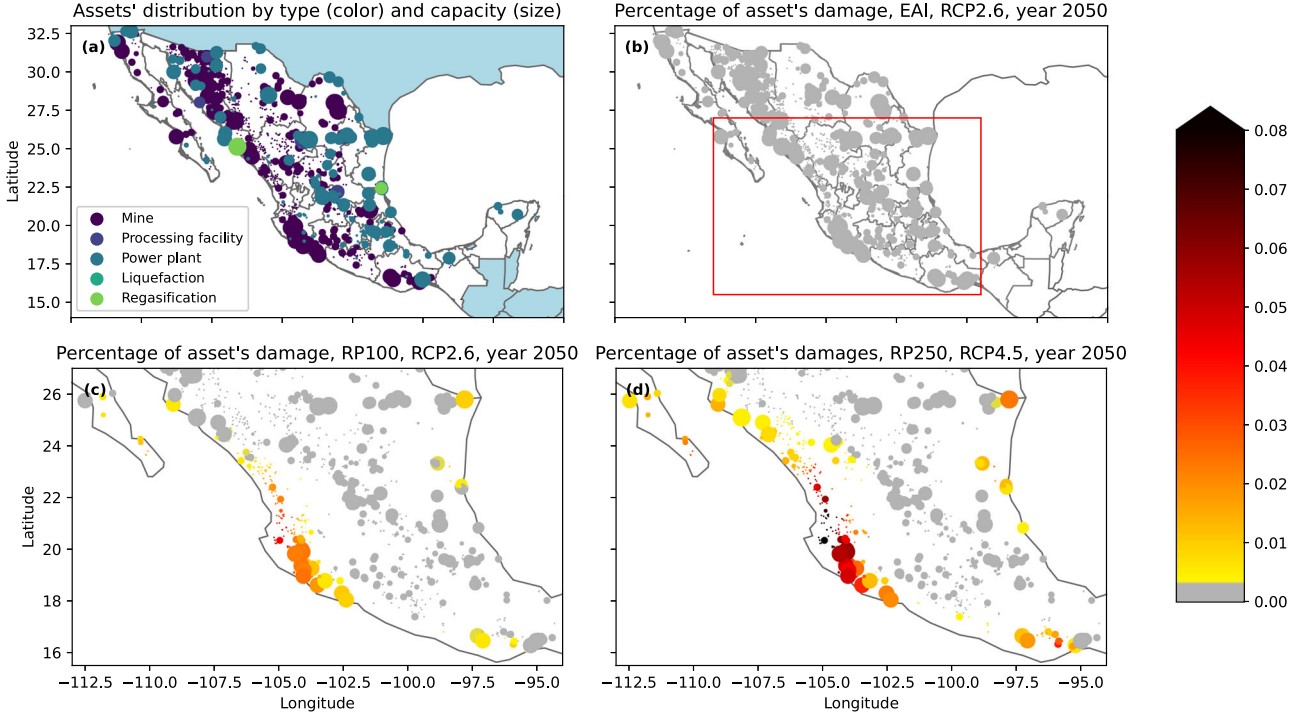

**Fig. 2 | Assets' distribution and direct impact of tropical cyclones on assets.** All panels: assets are represented as dots. The position of the dots is determined by the latitude and longitude coordinates of the asset. The size of the dots is proportional to the standardized capacity of the asset (i.e., assets with larger capacity will have larger dots). **a** Assets' distribution. The colour of the dot describes the type of asset (e.g., mine, power plant, see legend in the bottom left of the panel). **b** Percentage of direct damages from tropical cyclones on assets under Expected Annual Impacts (EAI), Representative Concentration Pathway 2.6 (RCP2.6), year 2050. As expected, all assets have a nearly zero direct impact under EAI (i.e., grey colour). The red rectangle delimits the area which is zoomed in (**c**, **d**). **c** Percentage of direct damages from tropical cyclones under a Return Period of 100 years (RP100), RCP2.6, year 2050. **d** Percentage of direct damages from tropical cyclones under a Return Period of 250 years (RP250), RCP4.5, year 2050. The colour bar to the right of the chart is common to (**b**–**d**) and relates the colour of the dot to the percentage of direct damages from tropical cyclones. Source: authors' elaboration on data from Refinitiv Eikon, S&P, and maps from geopandas (https://geopandas.org/en/stable/) and Open Street Maps (https://www.openstreetmap.org/about).

same time, neglecting the tail of acute risks leads to an underestimation of losses up to 97.7% (row 3, column 3).

**Neglecting asset-level information leads to a relevant underestimation of losses for investors**

In the absence of granular asset-level data, a possible approximation, which we refer to as using "proxy data", consists of replacing the set of locations where the firm operates by only one location, generally the country, city or postal code of its headquarter (see e.g., ref. 31). However, the impact of such approximation on the loss estimates is still unknown and deserves investigation. Hence, here we compare the portfolio losses computed using the two approaches: asset-level data vs proxy data. More in detail, for the same set of firms, we perform the equity valuation adjustments using asset-level data, and then using only the firms' headquarters as proxy for the location of its operations. By comparing the results, we determine the underestimation of losses resulting from neglecting asset-level data. We show that this loss of information has important implications for climate financial risk analysis. We illustrate this result in Table 2A, B. We first compare the losses on equity computed using the asset-level data to the losses computed using proxy data. We show that the magnitude of the VaR is 3 to 5.5 times larger when computed using asset-level data (Panel A, rows 2 and 6, third column) with respect to using proxy data (Panel A, rows 4 and 8, third column). As shown in Table 2B, neglecting asset-level data leads to an underestimation of the portfolio VaR between 67.4% and 92.3% when not considering chronic risks (Panel B, row 1, last column). The underestimation is lower when considering also chronic risks, but still in the range from 58.0 to 70.8% (Panel B, row 2, last column).

## Discussion

Losses from climate physical risks are expected to increase in the future due to growing climate change impacts, compounding of climate physical risks[32–34], and insufficient countries' mitigation efforts. Our methodology provides an assessment of financial losses from climate physical risks more robust than those previously available, by considering firms' asset-level information and business lines, and by combining acute and chronic physical risks scenarios. In turn, a better assessment of firms' exposure to physical risk can support advances in academic research in adaptation, as well as the design of policies and financial instruments aimed to build resilience to climate change and fill the adaptation gap.

The magnitude of financial shocks from climate physical risks can crucially depend on the geolocation of firms' assets and on their business lines. For instance, firms that operate in the same economic sector may have a different exposure to physical risk, and thus a different risk profile, due to their asset-level (climate and non-climate) characteristics. These factors drive the two main results of our paper. First, neglecting tail acute risks leads to a large understimation of portfolio losses, up to 82.2%, in our sample. Second, importantly, using proxy data instead of asset-level information, leads to a relative underestimation of the VaR of the investor up to 70.8%. Thus, asset-level information is key for climate physical risk assessment.

Our results are conditioned to the following limitations, some of which, however, apply more in general to the field of climate physical risk assessment. We acknowledge that our analysis does not take into account all the potential sources of uncertainty. Thus our results provide a quantification of losses conditional to the following specifications. First, we rely on point estimates of the characteristics of assets

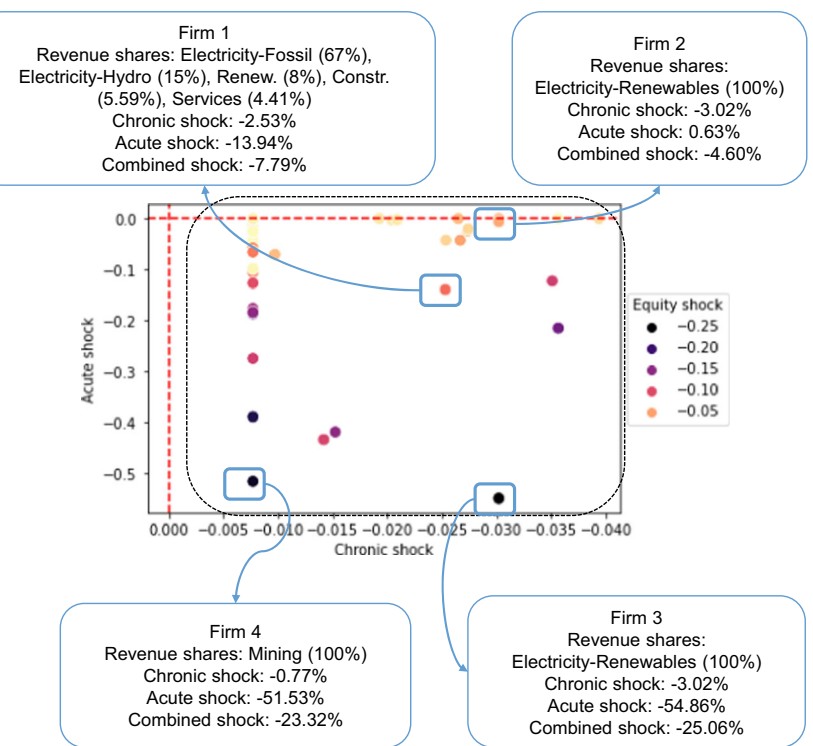

1. Diversified firms (Firm 1) have both acute and chronic shocks depending on share of revenues from assets and geolocations

2. Firms can have similar chronic shock (because same sector) but very different acute shock (due to geolocalization, Firm 2 vs. Firm 3)

3. Firms can have large acute shocks even if operating in different sectors (Firm 3 vs. Firm 4)

4. Firms can be affected by similar large acute shocks but different chronic shock (same pair as above)

**Firm 1**
Revenue shares: Electricity-Fossil (67%), Electricity-Hydro (15%), Renew. (8%), Constr. (5.59%), Services (4.41%)
Chronic shock: -2.53%
Acute shock: -13.94%
Combined shock: -7.79%

**Firm 2**
Revenue shares: Electricity-Renewables (100%)
Chronic shock: -3.02%
Acute shock: 0.63%
Combined shock: -4.60%

**Firm 4**
Revenue shares: Mining (100%)
Chronic shock: -0.77%
Acute shock: -51.53%
Combined shock: -23.32%

**Firm 3**
Revenue shares: Electricity-Renewables (100%)
Chronic shock: -3.02%
Acute shock: -54.86%
Combined shock: -25.06%

**Fig. 3 | Scatter plot showing, for each firm (dot), the acute, chronic and equity shocks.** Selected scenario: Shared Socioeconomic Pathway 3-Representative Concentration Pathway 4.5 (SSP3-RCP4.5), year 2040. Sample size: 86 firms with available asset-level data. X axis: chronic shock, defined as the loss in sectoral output for the business lines in which a given firm is engaged, weighted by the respective revenues. A negative number indicates a loss (e.g., −0.04 indicates a 4% chronic shock) and a positive number a gain. Y axis: acute shock on assets representing the asset-level losses borne by a firm, described by the average across all assets of $-\eta_a$, as defined in Eq. (9). A more negative value represents a higher shock and a value of 0 represents no losses on assets. Colour: equity shock, i.e., final outcome of the full Climate Dividend Discount Model (CDDM). A negative value (darker colour) represents a larger negative shock on equity valuation. The figure highlights four firms with different business lines, chronic, acute, and combined shocks. The left hand-side box discusses the implications of pairwise comparisons across these highlighted firms.

(e.g., residual life, monetary value, capacity) because confidence intervals on those estimates are not available. Second, we take the point estimate for RP and EAI of tropical cyclones impacts from the CLIMADA model under the standard setting[13]. Indeed, there is not yet an established way to calibrate the set-up that could yield uncertainty on the estimates of RP and EAI[35]. In addition, the data needed to properly calibrate assumptions about uncertainty is not available. Third, in order to quantify the impact of tropical cyclones, we relied on traditional approaches in the literature (e.g., refs. 21 and 35) that interpolate between current (2001–2020) and future (2081–2100) climate scenarios but they do not take into account the uncertainty on the evolution of the impacts. Fourth, we use the same damage function across all types of assets since data on damages by asset type are not available. In this regard, publicly available datasets, such as EM-DAT[37] only provide damages per event, at the country or (more rarely) at the subnational level. Finally, we do not consider firms' adaptation efforts (such as assets' relocation, implementation of physical barriers, etc.) because this information is currently not available[38]. Further limitations are discussed in "Methods". We provide an estimate of how uncertainty and adaptation efforts could affect our results in Supplementary Sections 10 and 11.

Our methodology can be applied beyond the Mexico case presented in this study. In fact, the presented steps can be tailored to assess physical risks for different countries (e.g., the US), hazards (e.g., floods) and securities (e.g., bonds, after tailoring the financial model). The main limitation to further applicability is represented by data availability. Our analysis has important implications for several types of decision-makers in climate finance. On the one hand, it contributes to inform policymakers in the design and implementation of

adaptation policies, in particular in the current context of public budget constraints and rising public debt, both in high and low-income countries. For example, in the EU, the European Climate Adaptation Strategy called for smarter adaptation and highlighted the need to translate large volumes of climate information into customized and user-friendly tools[39]. In this regard, our methodological framework sets a blueprint for the use of geolocalized climate data for climate finance analysis, and supports the Strategy's objectives, such as closing the adaptation gap, integrating climate risks and resilience into fiscal frameworks, and scaling up finance for climate resilience. In addition, our framework contributes to inform financial regulators and supervisory authorities to better assess the impact of climate physical risks for sovereigns, considering the composition of each country's economy (e.g., contribution of firms and sectors to Gross Value Added and fiscal revenues), the impact on sovereign risk (e.g., via bonds' yields and spreads) and the potential financial and regulatory response (e.g., monetary and prudential policies[40]).

On the other hand, our methodology supports financial institutions in avoiding underestimations of exposure to physical risks. In fact, while some financial actors such as insurers and reinsurers are relatively well-prepared in catastrophe risk management, others are less sophisticated[41]. This is for instance the case of banks' lending to firms (which represents on average 40% of EU banks' total assets, https://www.eba.europa.eu/risk-analysis-and-data/risk-dashboard) and of institutional investors (e.g., asset managers, pension funds and investment funds). Importantly, even a well-diversified portfolio would bear high physical risks as extreme events are predicted to increase globally[2]. Thus, geographic and sectoral diversification have limited benefits for portfolio climate risk reduction. In contrast, our

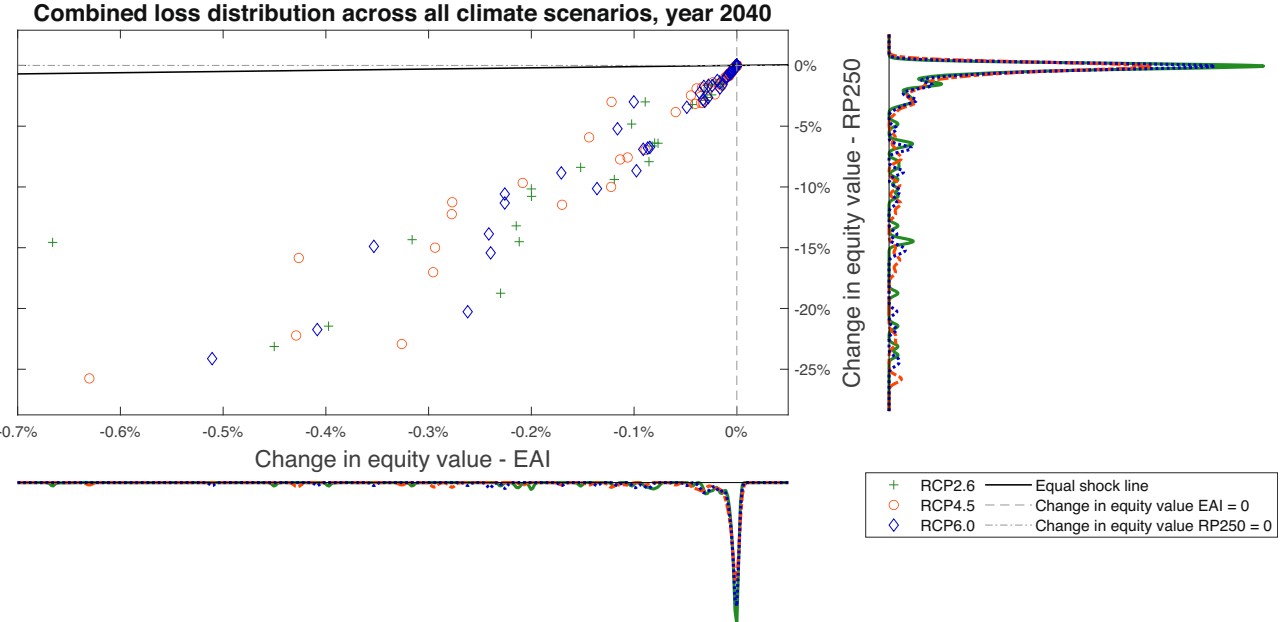

**Fig. 4 | Scatter plot for the joint Expected Annual Impacts (EAI) and 250-year Return Period (RP250) equity loss distributions for the year 2040, 86 firms with available asset-level data, considering acute risks only.** Scenarios are shown in different colours and dot styles, respectively: Representative Concentration Pathway 2.6 (RCP2.6; green plus), RCP4.5 (red circle), RCP6.0 (blue diamond). The marginal distributions for Expected Annual Impacts (EAI) and 250-year Return Period (RP250) are represented, respectively, on the bottom and right axis of the chart, approximated as kernel density plots for all scenarios and distinguished by the respective colours. The scatter plot represents the joint distribution. The loss is defined as $\frac{\tilde{V}_0 - V_0}{V_0}$, where $V_0$ is the equity value without the acute shock and $\tilde{V}_0$ is the shocked equity value. Thus, a negative value represents a loss (e.g., −4% represents a decrease in equity value of 4%). The black line labelled equal shock line represents the points where the loss conditioned to EAI is the same as the loss conditioned to RP250, implying no exposure to tail acute risks. Firms lie on the equal shock line (signalling no RP250 exposure) or below it (signalling high RP250 exposures). The dashed lines represent a 0% change in equity value under EAI (vertical line) and RP250 (horizontal line). Note that, for readability, the x axis left limit is set to −0.7%.

**Table 1 | Portfolio-level results, conditioned to Shared Socioeconomic Pathway 3-Representative Concentration Pathway 4.5 scenario (SSP3-RCP4.5), year 2040**

| Row | Case | Estimate (%) | Confidence interval (%) |
|---|---|---|---|
| **Panel A: portfolio-level results** | | | |
| 1 | Chronic only | −0.76 | (−0.98, −0.59) |
| 2 | EAI, asset-level (mean) | −0.085 | (−0.16, −0.049) |
| 3 | RP250, asset-level (VaR) | −3.3 | (−4.9, −2.2) |
| 4 | Chronic + EAI, asset-level (mean) | −0.84 | (−1.09, −0.66) |
| 5 | Chronic + RP250, asset-level (VaR) | −3.9 | (−5.5, −2.8) |
| **Panel B: underestimation of portfolio losses** | | | |
| Row | Compared cases | Underestimation range (%) | |
| 1 | Chronic vs tail acute (asset-level, RP250) | 73.2–79.3 | |
| 2 | Chronic vs chronic and tail acute (asset-level, RP250) | 78.8–82.2 | |
| 3 | Average acute (asset-level, EAI) vs tail acute (asset-level, RP250) | 96.7–97.7 | |

Panel A: portfolio-level results showing the mean and Value at Risk (VaR) computed for different cases of physical risk. The second column (Case) shows the selected case. The third column (Estimate (%)) shows the point estimate for the given metric and case. The fourth column (Confidence interval (%)) shows the 95% confidence intervals for the statistic in column 3, computed using the bias-corrected and accelerated percentile method over 15,000 samples. Cases labelled as asset-level are computed considering all data on assets for firms in the sample. Panel B: underestimation of portfolio losses, comparing cases pairwise. In each row, column 2 (Compared cases) lists the compared cases as case 1 vs case 2. Column 3 (Underestimation range (%)) is computed as the range of relative underestimation of the lower and upper bounds of the confidence intervals. The relative underestimation is computed as the relative difference between the boundaries of the confidence intervals for the first and second case. For example, in row 3 the underestimation range is computed as "[(confidence intervals, tail acute) – (confidence intervals, average acute)]./(confidence intervals, tail acute)", where "./" indicates element-wise division. Thus, the last column represents how large the underestimation of losses is when using case 1 instead of case 2: a value of 50% implies that using case 1 we fail to capture 50% of the risk as quantified using case 2.

methodology enables investors to better identify physical risks at the asset and firm level, considering assets' location. With this information, investment strategies, which may include a conditionality on adaptation investments from borrowing or invested firms, can be implemented, improving banks' and investors' resilience to physical risks. Finally, our analysis shows how firms could improve their climate risk management and facilitate physical risk disclosures under the

Corporate Sustainability Reporting Directive[42]. This is of key importance in light of the blind spots that still exist in firms' assessment of climate risks and in their adaptation strategies[38].

Furthermore, proper climate physical risk assessment matters for central banks and financial regulators with a financial stability mandate, to allow for the identification of policies and regulations aimed to mitigate systemic financial risk in their jurisdictions.

**Table 2 | Portfolio-level results, conditioned to Shared Socioeconomic Pathway 3-Representative Concentration Pathway 4.5 scenario (SSP3-RCP4.5), year 2040**

| Panel A: portfolio-level results | | | |
|---|---|---|---|
| Row | Case | Estimate (%) | Confidence interval (%) |
| 1 | EAI, asset-level (mean) | −0.085 | (−0.16, −0.049) |
| 2 | RP250, asset-level (VaR) | −3.3 | (−4.9, −2.2) |
| 3 | EAI, proxy (mean) | −0.013 | (−0.056, −0.0028) |
| 4 | RP250, proxy (VaR) | −0.59 | (−1.62, −0.17) |
| 5 | Chronic + EAI, asset-level (mean) | −0.84 | (−1.09, −0.66) |
| 6 | Chronic + RP250, asset-level (VaR) | −3.9 | (−5.5, −2.8) |
| 7 | Chronic + EAI, proxy (mean) | −0.77 | (−1.01, −0.61) |
| 8 | Chronic + RP250, proxy (VaR) | −1.3 | (−2.3, −0.8) |
| **Panel B: underestimation of portfolio losses** | | | |
| Row | Compared cases | | Underestimation range (%) |
| 1 | Tail acute (proxy, RP250) vs tail acute (asset-level, RP250) | | 67.4–92.3 |
| 2 | Chronic and tail acute (proxy, RP250) vs Chronic and tail acute (asset-level, RP250) | | 58.0–70.8 |

Panel A: portfolio-level results showing the mean and Value at Risk (VaR) computed for different cases of physical risk. The second column (Case) shows the selected case. The third column (Estimate (%)) shows the point estimate for the given metric and case. The fourth column (Confidence interval (%)) shows the 95% confidence intervals for the statistics, computed using the bias-corrected and accelerated percentile method over 15,000 samples. Cases labelled as asset-level are computed considering all data on assets for firms in the sample. Cases labelled as proxy are computed considering only proxy data for firms in the sample. Panel B: underestimation of portfolio losses, comparing cases pairwise. In each row, column 2 (Compared cases) lists the compared cases as case 1 vs case 2. Column 3 (Underestimation range (%)) is computed as the range of relative underestimation of the lower and upper bounds of the confidence intervals. The relative underestimation is computed as the relative difference between the boundaries of the confidence intervals for the first and second case. For example, on row 3 the underestimation range is computed as "[(confidence intervals, tail acute) - (confidence intervals, average acute)]./(confidence intervals, tail acute)", where "./" indicates element-wise division. Thus, the last column represents how large the underestimation of losses is when using case 1 with respect to case 2: a value of 50% implies that using case 1 we fail to capture 50% of the risk as quantified using case 2.

Finally, further investments in climate physical risk models are necessary to meet the needs of the financial sector[5]. At the same time, closer collaboration between climate modellers and economists is necessary to improve climate physical risk assessment at the asset, firm, and portfolio levels, ultimately enabling better investment and policy decisions. However, the quest for better models is no justification to delay action by investors and financial supervisors in assessing physical risks and climate change adaptation on the basis of available models.

## Methods
### Database model
We develop a database model to collect and logically connect extra financial, climate and financial information of individual physical assets and firms. The database model provides a granular and comprehensive overview of the characteristics of firms by collecting information on their productive assets, their business lines' composition and performance, and their financial and climate characteristic. We disaggregate firm-level information by asset and geography, considering the firm as a portfolio of business lines and geographically distributed assets[43]. This enables us to downscale climate risk assessment to the fundamental business units of the firm, considering their potentially different exposure to climate-related hazards, due to the geographical location of their productive assets.

First, we collect data on firms' revenues by business line. We leverage information on business units, product types, and their respective sales quantities and prices. Second, we retrieve, clean and consolidate a database of physical asset exposures from different data providers, for instance, Refinitiv Eikon (https://eikon.thomsonreuters.com/index.html), and S&P (https://www.spglobal.com/en/). As different databases provide different information on assets (e.g., some databases provide a monetary value, or a location, and some do not) as well as on owners (e.g., databases using different identifiers, or no identifiers but just firms' names), we set up a preprocessing pipeline to overcome these data fragmentation and comparability problems. Our process attaches to each asset a location, production capacity, monetary value, useful residual life, technology, operating status and ownership. We focus on energy-related or energy-intensive assets such

as power plants, liquified natural gas (LNG) facilities, and mines, given the relevance of physical risk for the energy sector (see ref. 44 for a comprehensive review) and its consideration as "sin stocks"[45]. Finally, we connect assets to business lines and thus firms' equity valuation.

Our combined databases return 123,340 physical assets globally. In total, 3493 are located in Mexico and we reconstruct their chain of ownership. In fact, asset-level datasets generally have information on direct owners, but not necessarily on the listed owners who issue financial contracts. To solve this issue, it is necessary to match owners' names and identifiers across multiple databases. By doing so, it is possible to reconstruct the chain of ownership from the asset to its listed owners and the issued financial contracts. See also Supplementary Section 1 for a list of used databases.

Ultimately, we link 1820 physical assets to 177 firms, both Mexican and internationally owned, that own the assets and are invested in by European financial actors. To these firms, we link 17,147 individual equity holdings of 1014 European investors consolidated via 199 different equity instruments. The total exposure value of European investors amounts to 290.11 billion USD (as of June 30, 2020).

### Climate physical risk assessment
Assessing physical risk on corporate securities differs from assessing physical risk for sovereign ratings or debt, especially in terms of data availability. In fact, country-level information on past disasters is available from publicly available databases (e.g., EM-DAT[37]), but historical firm-level information is missing. Thus, we rely on a probabilistic risk assessment approach to assess asset-level and firm-level losses. In addition, our methodology includes a dedicated treatment of the transmission channels and implications of climate physical risks on firms' business lines, on economic sectors and macroeconomic variables, the latter being captured with the use of a dedicated macroeconomic model (ICES).

### Asset-level assessment
We use the CLIMate ADApt (CLIMADA) model (https://wcr.ethz.ch/research/climada.html)[13,20,21,35,46] to perform a probabilistic assessment of damages from tropical cyclones at each asset location, for different Representative Concentration Pathways (RCP) scenarios (2.6, 4.5, 6.0)

at different years (2035, 2040, 2045, 2050). Supplementary Fig. 3 illustrates the workflow's schematic.

We consider historical data on tropical cyclones between 1950 and 2021 provided by the International Best Track Archive for Climate Stewardship (IBTrACS) (https://www.ncei.noaa.gov/products/international-best-track-archive), for the North Atlantic and Eastern North Pacific basins. Of 1555 historical events originated from either basin, 336 crossed Mexico. A map of these events is included in Supplementary Fig. 4. We standardize events' tracks by interpolating wind speeds at half-hour time steps. Building on Aznar-Siguan and Bresch[13], we simulate 50 synthetic tracks for each historical event, including track decay after landfall, for the probabilistic assessment. We remove duplicate hazards in the set to obtain a final dataset of 16,728 cyclones. Hazards can be duplicated in the first place as we are using two different basins as reference points, and tracks can cross from one another.

Tracks are then mapped to centroids in Mexico, i.e., geographical points where we define a wind speed from the track. The grid is set at 0.2 degrees of latitude/longitude, for a total of 14,076 centroids matched to hazards. We also tested the effect of using a finer grid on asset-level damages for the year 2040, see Supplementary Section 4 Asset-level damages with a finer grid. The comparison shows that using a finer grid could lead to slightly higher asset-level damages. We use CLIMADA to perturb the tracks for future climate change impacts for a given RCP scenario and year. The procedure followed in the model is based on the results obtained in ref. 47 for RCP4.5. Changes in tropical cyclones' frequencies and intensities are then obtained by linear interpolation for different RCPs and years. In this study, we use RCP2.6, 4.5 and 6.0, and years 2035, 2040, 2045 and 2050. The choice of RCPs and years is made to match the set-up of the ICES and CDDM models. For limitations of this procedure, see CLIMADA's documentation and references therein (https://climada-python.readthedocs.io/en/stable/tutorial/climada_hazard_TropCyclone.html).

By combining the wind speed at the centroid closest to a given asset and a damage function, we obtain asset-level impacts. The damage function describes the relation between the wind speed and the damages to a given asset. The formulation used is shown in Eq. (1)[48].

$$F_{index} = \frac{v^3}{1 + v^3},$$ (1)

where

$$v = \frac{max((W_{spd} - W_{thresh}),0)}{W_{half} - W_{thresh}},$$ (2)

Equation (1) enables the translation of wind speed ($W_{spd}$) into direct damages to assets described by the fraction of damaged property $F_{index}$ via a cubic power. It also considers a lower bound $W_{thresh}$ of no damage occurrence and a value $W_{half}$ where half the damage occurs. We follow the calibration provided in ref. 33 for Mexico and select $W_{thresh}$ to be 65 km/h and $W_{half}$ to be 253 km/h. Other calibrations exist in the literature, for example, ref. 49, which sets $W_{thresh}$ = 92.52 km/h and calibrates $W_{half}$ with two different approaches to either 214.56 km/h or 238.68 km/h. In comparison, the calibration in ref. 33 may overestimate the damages from low-category hurricanes and underestimate the damages from high-category hurricanes. Both the calibrations by Dunz et al.[33] and Eberenz et al.[49] are based on the shape of the damage function proposed by Emanuel[48]. However, the former calibration is performed on disaster damage data from Mexico only, while the latter is performed on disaster damage data from Mexico and the Caribbean (for $W_{half}$), and on disaster damage data from the US (for $W_{thresh}$, consistently with ref. 48). We use the calibration by Dunz et al.[33] for this study, as it is specific to Mexico only. The damage function in Eq. (1) considers only wind speed. This is a common assumption in the

literature[13,33,48]. Nevertheless, considering only wind speed limits the extent of the assessment of those hurricanes where rainfall and storm surges can account for high damage, despite the storm being less windy[13]. Importantly, we keep the damage function constant across asset types. Calibration of asset-level damage functions is left for further research.

We use two measures of damages at each scenario-time combination: Expected Annual Impacts (EAI) and 250 years Return Period (RP250). These combine the damage functions and the hazards to obtain measures of average (EAI) or tail (RP250) risks on assets. EAI is computed as:

$$EAI_j = \sum_{i=1}^{N_{ev}} x_{ij} F(E_i),$$ (3)

where $x_{ij}$ is the realization of the random variable $X$ representing the impact, index $j$ denotes a physical asset, $E_i$ is an event, $F$ its annual frequency and $N_{ev}$ is the number of (independent) events considered.

For cyclones, return periods are defined as "the frequency at which a certain intensity of a hurricane can be expected within a given distance of a given location" (https://www.nhc.noaa.gov/climo). For example, a return period of 20 years for a hurricane means that on average during the previous 100 years, a hurricane of a certain category or greater passed within 50 nautical miles (58 miles) of a given location about five times. Importantly, a 1-in-100-year event will not necessarily occur once in a century but may also occur more often, or not occur.

For more details on the estimation of return periods, see ref. 50. For the implementation in CLIMADA, the reader is referred to refs. 13 and 20 and the model documentation (https://climada-python.readthedocs.io/en/stable/index.html).

## Macroeconomic assessment

We use the ICES model (https://www.icesmodel.org/) to quantify macroeconomic impacts of chronic risks[23,24], as applied within the COACCH project (https://www.coacch.eu/)[22]. We source gross domestic product (GDP) and sectoral output trajectories under different combinations of scenarios (Shared Socioeconomic Pathways (SSPs) and RCPs), assumptions on capital mobility, and level of climate change impact. Trajectories are provided as "baselines", i.e., without climate change, and as "impact scenarios", i.e., including climate change. Thus the output change from a baseline to an impact scenario is dependent on climate change only. We use the following SSP-RCP combinations for our study, at the time horizons 2035, 2040, 2045, 2050: SSP2-RCP6.0, SSP3-RCP2.6, SSP3-RCP4.5, SSP5-RCP4.5. For the purpose of this study, we use the assumptions of high climate change impacts and high capital mobility in ICES. The ICES model as used in COACCH is resolved in 5-year steps until 2070, though for this study, we use a 2050 horizon. Further description of ICES' sectors and the scenario choice are provided in Supplementary Sections 2 and 5.

## Climate financial valuation: the Climate Dividend Discount Model

We quantify climate physical risk adjustments on equity valuation by developing a Climate Dividend Discount Model (CDDM). It extends the traditional Dividend Discount Model (DDM) framework[25,26] in its three stages formulation[28] to account for acute and chronic risks on firms' long-term growth. The former depends on assets and extreme events, the latter on business lines and their economic trajectories. To estimate the market value of equity, DDMs discount the future dividends using a discount rate to determine their present value. The discount rate represents the rate of return required by investors. Alternative formulations of this discounting concept exist, for example, based on Discounted Cash Flow (DCF). Importantly, our methodology can be applied to DCFs too. We assume the following:

- Dividends can be estimated by combining Earnings Per Share (EPS), their growth, payout ratios, and their respective long-run trends.
- Long-term climate physical risks are not accounted for in the current valuation[51]. Thus, the long-run growth rate used for equity valuation is not consistent with future climate risks. We assume this growth rate without climate impacts is constant across all firms.
- Physical risks are going to impact mostly the long-run part of the valuation.
- Discount rate is constant for all firms in all periods.

The equity value at time 0 ($V_0$) is computed as:

$$V_0 = \sum_{t=1}^{t_1} \frac{D_t}{(1+r)^t} + \sum_{t=t_1+1}^{t_2} \frac{D_t}{(1+r)^t} + \frac{V_{t_2}}{(1+r)^{t_2}} \tag{4}$$

where $D_t$ represents the dividend at time $t$, $r$ the discount rate, and $V_{t_2}$ the terminal value once the explicit estimation of dividends ends. $t_1$ and $t_2$ are the boundaries of the first and second stages. The following relation connects the dividends to Earnings Per Share (EPS):

$$D_t = EPS_t(1 - b_t), \tag{5}$$

where $EPS_t$ represents the earnings per share at time $t$, and $b_t$ the earnings retention rate making $(1 - b_t)$ the payout ratio. We obtain EPS and Dividends Per Share (DPS) from S&P. Data are available for a generally limited number of years. Thus, to complete the dividend series until $t_2$, we first estimate EPS as:

$$EPS_t = EPS_{t-1} g_{EPS,t,t-1}, \tag{6}$$

where $g_{EPS,t,t-1}$ represents the EPS growth rate between $t-1$ and $t$, and it holds $g_{EPS,t_2} = g_L$, where $g_L$ represents the long-run growth rate of dividends. Importantly, this relation implies a linear decline of EPS growth towards $g_L$. Thus, Eq. (6) enables the estimation of missing EPS, and hence dividends from Eq. (5). Then, the three stages in Eq. (4) can be distinguished as follows. First, a stage where dividends are explicitly estimated by a data provider (for this study, S&P). Second, a stage where dividends are modelled with linear decline. Third, the estimation of a terminal value.

We can rewrite Eq. (4) for each firm as:

$$V_{0,j} = \sum_{t=1}^{t_{j,1}} \frac{D_t^j}{(1+r)^t} + \sum_{t=t_{j,1}+1}^{t_{j,2}} \frac{D_t^j}{(1+r)^t} + \frac{D_{t_{j,2}}^j(1+g_L)}{(1+r)^{t_{j,2}}(r - g_L)}. \tag{7}$$

where $g_L$ represents the long-term growth rate of dividends, i.e., the rate at which the firm reaches an equilibrium where investment opportunities, on average, earn their opportunity cost of capital. The index $j$ represents the $j$th firm. Differently from Eq. (4), we now make the dependence on $j$ explicit, a necessary step for practical applications. In fact, dividend data are generally sourced from a provider whose analysts will perform the in-depth analysis necessary for dividends' estimation. Depending on the firm, analysts will use different assumptions while modelling dividends. The usage of firm-specific assumptions for the estimation is reflected in the formula by the $j$ subscript, which shows how dividends are going to differ from firm to firm (denoted $D_t^j$) and how the boundaries of the stages are not fixed a priori but depend on the firm (denoted by $t_{j,1}$ and $t_{j,2}$). Note that dividends are explicitly modelled in the first stage, up to $t_{j,1}$ and subsequently reverted to long-run growth rates. This explicit step is necessary to link the theory of equity valuation to its practical implementation. For a discussion of the sensitivities of the model to $r$

and $g_L$, please see Supplementary Section 7. For this study, we set $r = 0.09$ and $g_L = 0.06$.

Due to climate change, we cannot keep the long-run growth constant across firms as in e.g., Refinitiv's StarMine model (https://www.lseg.com/en/data-analytics/financial-data/analytics/quantitative-analytics/starmine-intrinsic-valuation-model), as firms will be heterogeneously impacted. The growth rate of a firm will ultimately depend on: the output trajectories of its business lines as impacted by chronic risks, and the location and characteristics of its assets as impacted by acute risks. Importantly, our results shall be considered conservative due to the complexity of asset-level data, the compensation coming from positive macroeconomic shocks, and the considerations of only tropical cyclones.

We can now define the long-run growth rate as adjusted by physical risk considerations, $\tilde{g}_L$ as:

$$\tilde{g}_{L,(I,j)} = g_L \sum_{i=1}^{K_j} \left[ \frac{O_{i,I}}{O_{i,B}} \frac{1}{\delta_{j,I,i}} s_i \right], \tag{8}$$

where $I$ denotes a climate change impact scenario, $O_{i,B}$ and $O_{i,I}$ the output trajectories for sector $i$ under scenarios $I$, impacted by climate change, and $B$, without climate change. We define as chronic shock the ratio $\frac{O_{i,I}}{O_{i,B}} - 1$, meaning for instance that a loss on output, relative to the baseline, of 5% corresponds to a shock of −0.05. Note that, when entering Eq. (7), $\tilde{g}_L$ is always discounted by $(1+r)^{t_2}$, regardless of the considered damage measure for $\delta_{j,I,i}$. This implies that, when considering the RP250 scenario, the discounting for changes in $g_L$ is still $(1+r)^{t_2}$ and not $(1+r)^{250}$. The parameter $\delta_{j,I,i}$ depends on the firm- and business line-specific loss due to acute risk conditioned to scenario $I$ for sector (business line) $i$; $s_i$ is the applicable revenue share for business line $i$, and $K_j$ represents the total number of business lines for firm $j$. We design $\delta_{j,i}$ as a firm- and business line-specific variable computed as an average of all $\delta_{a,j,i}$, i.e., the impact for all physical assets $a$ owned by firm $j$ contributing to business line $i$. In our application, $\delta_{a,j,i}$ depends on three main parameters for the asset: its monetary value, its residual useful life, and the impact from tropical cyclones computed in CLIMADA.

$$\delta_{a,j,i} = 1 + \eta_a = 1 + \frac{L_a}{V_a} f(R_a) \tag{9}$$

where $R_a$ is the residual life of asset $a$, $f(R_a)$ a coefficient proportional to the residual life (for the purpose of this application, $f(R_a) = \frac{1}{\tau} R_a$, where $\tau$ equals 1 year), $V_a$ is the value of asset $a$, $L_a$ is the impact on asset $a$. Thus, $\eta_a$ represents an estimate of the relative impact (i.e., fraction of the asset value) on assets from tropical cyclones, taking into account the residual life of the asset. Importantly, $\delta_j$ is floored to 1 and capped to 2, i.e., we assume that firms do not benefit from having physical assets less exposed to climate physical risk, hence simply follow the general sectoral trajectories of their business lines. This assumption is made for simplicity, since assessing the existence of positive effects stemming from asset location requires an analysis which is beyond the scope of the paper. Note that the estimate of $\delta_{j,I,i}$ is computed on available assets but applied to the full business line.

Combining Eqs. (7) and (8) we obtain the CDDM formulation for the adjusted equity value ($\tilde{V}_{0,I,j}$):

$$\tilde{V}_{0,I,j} = \sum_{t=1}^{t_{j,1}} \frac{D_t^j}{(1+r)^t} + \sum_{t=t_{j,1}+1}^{t_{j,2}} \frac{D_t^j}{(1+r)^t} + \frac{D_{t_{j,2}}^j(1+\tilde{g}_{L,I,j})}{(1+r)^{t_{j,2}}(r - \tilde{g}_{L,I,j})}. \tag{10}$$

Where $\tilde{g}_{L,I,j}$ is the adjusted growth rate as per Eq. (8) and $\tilde{V}_{0,I,j}$ is the adjusted equity valuation conditioned to a given impact scenario $I$. Thus the combined equity shock, stemming from the revaluation of

shares considering chronic and acute physical risks, is given by:

$$\psi_j = \frac{\tilde{V}_{0,j,I} - V_{0,j}}{V_{0,j}}. \tag{11}$$

To interpret the equity shock we proceed as follows. The asset-level impact from tropical cyclones is a random variable $I_a$. Characterizations of its distribution in terms of EAI and RP are provided by CLIMADA. Since $I_a$ is a random variable, then $\tilde{g}_L$, $\tilde{V}_{0,j,I}$ and $\psi_j$, being functions of $I_a$ are also random variables and the distributions of their values could be generated from the distribution of the values of $I_a$.

However, this is a computationally expensive procedure. For the purpose of this paper, we extract selected moments and quantiles of the distribution of $I_a$, and compute $\tilde{g}_L$, $\tilde{V}_{0,j,I}$ and $\psi_j$ only for those moments and quantiles. Specifically, we select the first moment, i.e., the expected annual impact (EAI) and the 99.6th quantile (RP250). Note that with regard to the estimation of Value at Risk, since the loss on equity valuation is an increasing function of the economic loss, the quantiles of the adjusted equity valuation ($\tilde{V}_{0,j,I}$) are equivalent to the adjusted valuation computed on the quantiles. In contrast, regarding the estimation of expected annual impact, we approximate the average of the adjusted equity valuation ($\tilde{V}_{0,j,I}$) with the adjusted valuation computed on the average of the impact ($I_a$). Hence, we assume that using EAI for the equity valuation leads to computing the average adjusted valuation. Similarly, we assume that using RP250 for the equity valuation leads to computing the Value at Risk (VaR) of a firm's adjusted valuation. As such, for firms, the equity value computed using EAI represents an average equity value from the distribution of possible values considering physical risks. Similarly, the equity value computed using RP250 represents a percentile, or a VaR, of the distribution of possible equity values considering physical risks. Thus, the interpretation of the adjusted equity value is as follows. The firm has multiple possible growth paths in the long run. The equity value computed using RP250 corresponds to a future where the firm suffers damages from tropical cyclones that are comparable in magnitude to the ones emerging from an RP250 hurricane conditioned to a given climate scenario. Similarly, the equity value computed using EAI corresponds to a future where the firm suffers damages from tropical cyclones that are comparable in magnitude to the yearly expected damages. Importantly, also $O_{i,I}$ is a realization of a random variable and we can interpret it to be an average chronic risk impact. Analysing the relation between the two random variables is out of the scope of this paper. As such, we treat the realization $O_{i,j}$ as an average of chronic risks. We assume the equity valuation computed with the product of $O_{i,j}$ and average acute risks (EAI) approximates the average equity valuation considering physical risks. Similarly, we treat the combination of the realization of $O_{i,I}$ and RP250 as a (tail) percentile. The interpretation is the same also in the presence of chronic risks: firms are supposed to be exposed to effects corresponding to a given hurricane and to chronic effects as described by $O_{i,I}$, conditioned to a given climate scenario. Also, the adjusted equity value computed combining chronic risk and EAI still represents an average value, and the adjusted equity value computed combining chronic and RP250 still represents a VaR.

Importantly, in Eq. (8), $\delta_{j,I,i}$ is computed using the relative damages to assets as calculated by the CLIMADA model. Thus, in our model the impact of acute shocks on the firm is captured in a reduced form as adjustment in the growth rate of the firm. In this treatment, the ratio of asset damages (as computed in Eq. (1) using the damage function) links acute risks to the growth rate. Hence, it is not necessary to model the growth of assets explicitly. This approach is also consistent with the one followed in the macroeconomic model ICES. In fact, ICES represents the impacts of climate change either as changes in productivity or as losses on physical capital and land. Thus, the focus on relative losses to assets is consistent with the treatment of physical capital in the macroeconomic model.

The CDDM is computed conditioned to the following scenario combinations: SSP2-RCP6.0, SSP3-RCP2.6, SSP3-RCP4.5, SSP5-RCP4.5. In our model, the computation of the value of the firm takes into account the year span from 2022 to 2050. We assume firms are subject to climate impacts from 2035 onward, the period which captures the long run in the current treatment of the model. To proxy these impacts, we use a reference year for both the ICES model and tropical cyclones, namely 2040 (climate models' estimates at years 2035, 2040, 2045, 2050 are not distinguishable anyway in statistical sense[47]). In particular, the estimate of tropical cyclones' impacts at 2040 is obtained following a common approach in the literature based on a linear approach interpolation of the impacts between 2020 and 2100 (see e.g., refs. 21 and 36). Thus, for the valuation conditioned to e.g., scenario SSP2-RPC6.0, year 2040, ICES data are considered for SSP2-RCP6.0, year 2040, and tropical cyclones impacts are considered for RCP6.0, year 2040. Other years are not considered. Finally using a certain year to compute $g_L$ does not imply extending the dividend stream until that year, but only computing Eq. (8) with values for $O_{j,I}$ and $\delta_{j,I,i}$ for that year.

The full CDDM is applied to all firms with at least two datapoints available for EPS. For non-dividend paying stocks or stocks with missing data, we compute the equity shock as follows:

- For stocks paying no dividend, or with no dividend data, we revert to direct shocks, i.e., $\psi_j = \tilde{g}_{L,I,j} - g_L$.
- For stocks with data available only for the first period, we compute a one period version of the CDDM following $\tilde{V}_{0,j} = \frac{D_0}{r - \tilde{g}_{L,I,j}}$.

## Analysis using proxy data

We compare the portfolio-level results for losses computed using proxy data vs asset-level data. The purpose of this analysis is to quantify the relevance of the underestimation of physical risk that stems from neglecting asset-level data. To compute the results with proxy data, we replace asset-level data with one location per firm and use information on physical risk at this location to proxy physical risks for the firm. Thus, the CDDM model is computed for the same firms both using asset-level data and using only one location (i.e., proxy data). Business lines are used for comparability, i.e., the valuation is applied only to those business lines that are adjusted in the asset-level analysis. The single location is either the headquarter (for Mexican firms), the address of the Mexican subsidiary with highest ownership (for non-Mexican firms with Mexican subsidiaries) or Mexico City (for non-Mexican firms without Mexican subsidiaries). All addresses are geolocalised using Opencage API (https://opencagedata.com/) and checked manually.

For the one location, the value exposed to acute physical risks is given either by the firm's Property, plant and equipment (PPE) (Variable "property plant & equipment−net total", sourced from Refinitiv Eikon) (if larger than 1 million USD) or by its Total assets (TA) (Variable "total assets", sourced from Refinitiv Eikon). Using the methodology described in Subsection Asset-level assessment, we assess impacts from CLIMADA on PPE or TA at the single location. We combine the losses from tropical cyclones with chronic risks computed from ICES and plug them in the CDDM. For comparability purposes, we use $\delta$ only on those business lines which can be impacted at the asset level. Otherwise, applying $\delta$ to business lines that cannot be impacted would impair the comparability of the counterfactual analysis. Subsequent statistics (e.g., VaR) are computed as in the version of the model using all asset information.

## Limitations

The following remarks complement the limitations acknowledged in the Discussion. First, we account for uncertainties on financial portfolio loss (VaR, mean) estimating the confidence intervals (CI) using bootstrapping (see e.g., ref. 30). Including the sources of uncertainties mentioned in the Discussion would likely lead to larger CI. Second, the reader should be aware that there is considerable uncertainty regarding the effects of climate change on the frequency and intensity of tropical cyclones for the middle of the century

especially at less-than continental scale. The methodology we applied here to quantify damages around the middle of the century builds on relevant literature in the field and relies on interpolation (e.g., refs. 21 and 36). Third, we do not consider firms' adaptation measures (such as sea barriers or mangroves) due to lack of data on firms' adaptation strategies[38]. Adaptation may vary across firms, and different firms may follow different schedules to implement adaptation measures. Moreover, many adaptation measures generally considered in the literature (e.g., mangroves for coastal protection, as in ref. 36) are not relevant for the types of assets that we analyse here. Similarly, relocation is not feasible for most of the assets we consider (e.g., mines or power plants must be located where natural resources are located). Existing calibrations of adaptation measures are either based on assumptions (e.g., ref. 36 in the case of the effect of mangroves on tropical cyclones' winds) or specific to individual countries and thus not applicable to Mexico (e.g., ref. 21). Furthermore, Mexico invests very little in adaptation[52,53]. Fourth, information on assets' location, ownership, value and residual life is often missing and has to be estimated. Moreover, some non-core firms' assets (e.g., deposits, warehouses) may be unknown even for firms where asset-level data are available. Furthermore, for some assets it is not possible to reconstruct ownership chains, or the unlisted nature of some of the owners makes the link to equity financial portfolios not possible. Fifth, in our assessment, we consider only one country (Mexico), one hazard (tropical cyclones), a selection of asset types (mostly energy-related), and one financial asset class (equities). Thus, our results in terms of financial risk for investors are conservative. Sixth, short-term risks are generally downplayed both in the macroeconomic model framework used (see ref. 23 for a discussion), and in the CDDM. Finally, we consider only direct impacts of tropical cyclones, and not their indirect ones such as supply chain disruptions, damage to infrastructure other than the assets in the sample, or loss of lives.

## Data availability
The data that support the findings of this study are available from S&P and Refinitiv Eikon, but restrictions apply to the shareability of these data, which were used under license for the current study, and so are not publicly available. Data are however available from the authors upon justified request subject to licensing agreements with S&P and Refinitiv Eikon.

## Code availability
The codes that support the findings of this study can be made available upon request for the purpose of academic research.

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

## Acknowledgements

G.B., A.Đ., and I.M. acknowledge the financial support of the European Commission H2020-funded CASCADES (CAScading Climate risks: towards ADaptive and Resilient European Societies) project, number 821010. All authors are very grateful to Adrian von Jagow (WU Wien) for support with initial data work; Chahan Kropf (ETH Zurich) for discussion about the implementation of CLIMADA; Nepomuk Dunz (the World Bank), Marie Briere (Amundi) and Stefano Ramelli (St Gallen University), the participants of the 15th International Financial Risk Forum 2022 and of the GRASFI conference 2022 for their useful comments on the initial draft. All authors are particularly grateful to Francesco Bosello, Elisa Delpiazzo and Ramiro Parrado (Euro - Mediterranean Centre on Climate Change (CMCC), and Ca' Foscari University of Venice) for their comments and support in implementing the results of the ICES macro-economic model and COACCH project's results into the analysis. The usual disclaimer applies.

## Author contributions

The authors contributed equally to the conceptualization of the manuscript. I.M., S.B., and G.B. led the writing of the manuscript. A.Đ. developed the database model. G.B. wrote the computer code.

## Competing interests

The authors declare no competing interests.
