## [Peer Review File · Nature Communications]

Asset-level assessment of climate physical risk matters for adaptation financeREVIEWER COMMENTS

Reviewer #1 (Remarks to the Author):

Thanks for a timely and relevant contribution to the literature. The paper is well written and the Mexico case provides a good illustration, indeed. A big limitation is the use of one single impact function, irrespective of the nature of the exposure, neither whether direct (assets) or indirect impacts (revenue) are dealt with. This should be stressed more in the Discussion(3) as few will read the Limitations section (5, where this is revealed) - and one sentence should be spent on this in the abstract, too. The current mention of "...smoothing modelling assumptions regarding e.g. the use of a common damage function for all assets" (page 13) does rather hide than reveal this.

As for prior art, not in the academic literature, but in essence carrying out the same analysis, you might consider to reference the following two reports:

S&P, 2015: The Heat Is On: How Climate Change Can Impact Sovereign Ratings,
<https://www.spglobal.com/ratings/en/research/articles/151125-the-heat-is-on-how-climate-change-can-impact-sovereign-ratings-9425836>

S&P, 2015: Storm Alert: Natural Disasters Can Damage Sovereign Creditworthiness,
<https://www.spglobal.com/ratings/en/research/articles/150910-storm-alert-natural-disasters-can-damage-sovereign-creditworthiness-9327571> (unfortunately both not freely available, but you can register at www.spglobal.com for free)

Some further comments (as line numbering was missing, I've quoted the sentence my comment relates to):

Page 2, 2nd last line: "...estimated in USD 250 billion per year,...", might better be phrased ...estimated to amount to...

Page 6, Figure 2, caption: "In the central regions of the country, assets are mostly not impacted (i.e. grey colour) as these regions are generally not interested by tropical cyclones.", you mean exposed, not interested, I presume. But what about torrential rain and flooding associated with tropical cyclones? In general, shorten the figure caption, as quite

some text is redundant (e.g. “The location of the dot is determined by the latitude and longitude coordinates of the asset.” three times).

Page 8, Figure 3: It is somewhat troubling to denominate acute shocks with positive numbers. Why did authors not denote these shocks as negative, as they did for chronic and combined shocks?

Page 9, Figure 4: “Scatter plot for the joint EAI and RP250 equity loss distributions for the year 2040, 86 companies with available asset-level data”. Are asset values scaled to 2040? How? See also my comment iro Page 14.

Page 13: With respect to implications:

1) “...contributes to inform policymakers in the design and implementation of adaptation policies...” can you be more specific in that sense?

2) “..supports financial institutions, such as insurance and reinsurance companies..”. This might be true for banks (see Westcott, M., Ward, J., Surminski, S., Sayers, P., Bresch, D.N. and Claire, B., 2020. Be prepared: Exploring future climate-related risk for residential and commercial real estate portfolios. *The Journal of Alternative Investments*, 23(1), pp. 24-34. <https://jai.pm-research.com/content/early/2020/05/09/jai.2020.1.100.abstract>), but insurers and reinsurers in particular do apply natural catastrophe models to price their business and manage their capital already. As the latter use risk models with bespoke impact functions (much more detailed than as used in the present study), one would not expect them to benefit much from insights as presented here. But in a wider TCFD context, there are many private sector actors (real economy, not only financial sector) who would benefit a lot from such disclosure, not least investors... yet full transparency is needed, which means the models along the whole chain as presented in the paper need to be open-source and free-access.

Page 14: With respect to asset data, how are future projections handled? I did not find a description of the scaling of assets (and related turnover or revenues) with SSPs (they are mentioned further below in the context of the macroeconomic assessment).

Page 15: “The grid is set at 0.2 degrees of latitude/longitude, for a total of 14,076 centroids matched to hazards.” There would be pre-computed probabilistic tropical cyclone hazard sets (today and RCPs) at a resolution of 150 arcsec (4 km) available via the CLIMADA data API (<https://climada.ethz.ch/data-types/>). Why did the authors resort to a coarser resolution of 0.2 degrees?

Page 16: “We follow the calibration provided by (Dunz et al., 2021) for Mexico.” Would be worth comparing with the parameters as determined by Eberenz, S., Lüthi, S., and Bresch, D. N., 2021: Regional tropical cyclone impact functions for globally consistent risk assessments, *Nat. Hazards Earth Syst. Sci.*, 21, 393-415, <https://doi.org/10.5194/nhess-21-393-2021>.

Page 16: “Importantly, we keep the damage function constant across asset types.”. This is a rather crude assumption, as authors state “Calibration of asset-level damage functions is left for further research”. It would be worth checking for stability of the results using variants of impact functions. The calibration as stated is based on damage to general property, while industrial assets usually show lower a loss burden. Business interruption (or revenue loss) is yet another story. Hence a proper sensitivity analysis might be considered, see Kropf, C., Ciullo, A., Otth, L., Meiler, S., Rana, A., Schmid, E., McCaughey, J. W., and Bresch, D. N., 2022: Uncertainty and sensitivity analysis for probabilistic weather and climate-risk modelling: an implementation in CLIMADA v.3.1.0. *Geosci. Model Dev.*, 15, 7177-7201, <https://doi.org/10.5194/gmd-15-7177-2022>

Page 22: “To perform the firm-level analysis, we replace asset-level data with one location per company and use information on physical risk at this location to proxy physical risks for the company.” Yet you stated on page 3 “...previous works on risk assessment have resorted to approximating the location of economic activities of each firm with its headquarter location”. Could you please explain why you deem a single point a good proxy for the asset portfolio (distributed across Mexico - and beyond) for a company?

Reviewer #2 (Remarks to the Author):

The authors clearly put in a lot of effort to integrate several databases in order to be able to

assess financial losses when neglecting asset-level information and acute risks. I have several issues with the study though. Most important:

1. The authors make the claim that neglecting asset-information and acute risks results in an underestimate of investor losses due to climate risks. I do not find this claim particularly interesting as it is true by definition. More interesting would be the question whether there is any evidence that investors (or policy makers or insurance companies) are actually underestimating these risks. Unfortunately the study is silent on this.

2. The authors use granular plant level data to construct firm level exposure to climate risk and claim to be the first to do this. They focus on Mexican firms and in particular risks related to cyclone damage. The authors do not seem to be aware of Moody's 427 measure of climate stress exposure for (more economically relevant) S&P500. This measure, similar to this paper, is based on the locations of physical assets such as production plants and offices which is then aggregated to the company level. Moody's 427 provides risk scores in several risk categories: heat stress, water stress (drought), flood (corporations only), extreme rainfall (municipalities only), hurricane, and sea level rise. While the authors of this study focus on cyclone risk, Acharya et al. (2022) convincingly show that investors are concerned with heat stress but do not seem overly concerned with the other risk categories (including cyclone risk).

Acharya, V. V., Johnson, T., Sundaresan, S., & Tomunen, T. (2022). Is physical climate risk priced? Evidence from regional variation in exposure to heat stress (No. w30445). National Bureau of Economic Research.

Reviewer #3 (Remarks to the Author):

This article seeks to provide a methodology for the quantification of physical climate risks and a procedure for translating these risks to financial losses at the company and portfolio level.

The article illustrates this methodology, and the procedures involved, through a sample of listed firms and activities located in Mexico, by assessing their exposure to (and the financial consequences of) tropical cyclones (TCs). It does so in five steps. First, through the

construction of an asset-level database that connects financial information at the level of a firms' business lines to physical asset exposures. Second, by using the CLIMADA model to perform probabilistic assessment of damages from TCs at each asset location for different Representative Concentration Pathways (RCP2.6, 4.5, 6.0) at different time periods (2035, 2040, 2045, 2050). Third, by connecting these acute impacts with sector-level chronic impacts, as computed using the ICES model. Fourth, by developing a Climate Dividend Discount Model (CDDM) that connects these asset-level impacts on a company, to provide a climate-adjusted stock value. Fifth, by translating these impacts to the investor through the calculation of portfolio Value at Risk (VaR).

Through the above, the authors assert investor losses are underestimated by up to 70%, when asset-level information is neglected, and by up to 82%, when acute risks are neglected.

The problems the authors are trying to solve for, notably the assessment of asset-scale physical climate risk at the portfolio-level, is one that is critical to financial system actors and one that the investor community is grappling with. More broadly, the authors refer to challenges in assessing the impact of climate physical risks at the scale of an asset, arising from the computing limitations of current generation climate models and model uncertainty. The authors are to be commended for their work in attempting to solve the problem of portfolio-level risk assessment and the challenges of incorporating asset-level data.

My chief concerns arise with regard to these challenges, as well as from an understanding of what constitutes climate risk. In the main, these concerns the first and second parts of the five-step framework developed. It is to these that I will speak to as this is where I am more qualified.

Asset-level risk (for extremes)

The approach to incorporating asset-level data into portfolio-level analysis is to my understanding novel and to be commended. The authors have, with some stated limitations (including incomplete data and estimations based on these), built a database of physical asset information, including location, production capacity, monetary value, useful residual

life, technology, operating status and ownership. In addition, the authors have linked this data to probabilistic assessments of future acute (hurricane) and chronic risks, to assess the exposure of these assets.

For the purpose of climate risk assessment, in particular risk assessment for extremes, risk is comprised of three elements: hazard, exposure and vulnerability (Cardona et al., 2012). While the hazard (hurricanes) and exposure (assets located in the path of the hurricane) are incorporated into the methodology developed, vulnerability (engineering of individual assets to particular standards, elevation, land surface type, tidal behaviour etc.) has to my understanding not. The relative vulnerability of a particular asset can, however, be instrumental in determining its preparedness for a particular hazard and, in this instance, its ability to adapt to hurricanes of greater intensity and frequency. Some assets may, for example, already be regularly exposed to TC risk and will accordingly be located at an elevation, on a land surface type and to a standard that renders them less vulnerable than others. I do not, for one instance, suggest that this problem is easy to overcome but would suggest, at the very least, that this important limitation be stated.

The probabilistic assessment of damages from tropical cyclones

The authors undertake a probabilistic assessment of damages from TCs based on Aznar-Siguan & Bresch (2019), which includes the introduction of synthetic cyclone tracks to historical cyclone tracks to overcome the problem of such events occurring only infrequently. These tracks are then perturbed in CLIMADA, as described in Knutson et al (2015), through the dynamical downscaling of global climate projections across RCPs 2.5, 4.5 and 6.0 for the years 2035, 2040 and 2050.

There are for me several issues that I will address each in turn.

- Timescale. In contrast to the work of Knutson et al (2015), who run simulations for present day (2001-20) and late twenty-first century (2081-2100), the authors attempt to assess changes in TC frequency and intensity across the years 2035, 2040 and 2050. As noted in Knutson et al (2015), however, changes in TC frequency and intensity at decadal scales cannot be determined because natural variability (i.e., influences on weather events from

phenomena like the El Nino Southern Oscillation) is not well understood, even if the longer-term outputs of the average changes of these is. For this reason, most climate projections across different RCPs tend not to be distinguishable until around 2050 onwards. This raises questions for the results illustrated in Tables I II for 2040.

- **Weather.** As also noted in Knutson et al (2015), as well as by CLIMADA (2017), the influence of local features and conditions such as cloud cover, topography, sea-surface temperature and humidity mean extreme events such as TCs develop at a scale not well represented in global climate model (GCM) projections, which CLIMADA data are based on. Accordingly, “how hurricane damage changes with climate remains challenging to assess” (CLIMADA, 2017). This similarly raises questions for how TCs will actually develop for the year 2040.
- **Uncertainty.** Any probabilistic assessment of a rare event based on global climate model simulations must include uncertainty bounds. My reading of the paper suggests confidence intervals at the 95th percentile are provided for the bootstrapping of average portfolio loss and VaR. However, it is not clear to me that uncertainties relating to the underlying GCMs and the likelihood of actual TC impacts (as discussed in the prior two points) propagates through into the estimation of investor losses illustrated in Tables I and II.

Advice with respect to publication

In conclusion, the results of this paper are without a doubt of interest and possibly indicative. My suggestion for publication would be for the authors to amplify the limitations and uncertainties arising from the confounding variables both the climate and the tools of climate science introduce, however.

Specifically, a significant contribution could be made to the literature, but also to policy-making more broadly, if the opportunity were taken in this paper to discuss the limitations of using climate science in its current form for the purpose of simulating the physical effects of extreme events at less-than continental-scales and for the years 2030-2050 in particular. Considerable advances and investment in climate modelling, as well as in co-operation between the climate modelling and economic communities, are needed if we are to develop the capacity to understand the financial effects of climate change for portfolios at the scale of a single asset more reliably. There is significant research potential in such collaborative work for both economics and science, and this article in this journal could have the effect of

motivating such work.

References cited

Aznar-Siguan, G., & Bresch, D. N. (2019). CLIMADA v1: a global weather and climate risk assessment platform. *Geoscientific Model Development*, 12(7), 3085–3097.

<https://doi.org/10.5194/gmd-12-3085-2019>

Cardona, O.-D., van Aalst, M. K., Birkmann, J., Fordham, M., McGregor, G., Perez, R., Pulwarty, R. S., Schipper, E. L. F., Sinh, B. T., Décamps, H., Keim, M., Davis, I., Ebi, K. L., Lavell, A., Mechler, R., Murray, V., Pelling, M., Pohl, J., Smith, A.-O., & Thomalla, F. (2012).

Determinants of Risk: Exposure and Vulnerability. In *Managing the Risks of Extreme Events and Disasters to Advance Climate Change Adaptation* (pp. 65–108). Cambridge University Press. <https://doi.org/10.1017/CBO9781139177245.005>

CLIMADA. (2017). Hazard: Tropical cyclones. https://climada-python.readthedocs.io/en/stable/tutorial/climada_hazard_TropCyclone.html

Knutson, T. R., Sirutis, J. J., Zhao, M., Tuleya, R. E., Bender, M., Vecchi, G. A., Villarini, G., & Chavas, D. (2015). Global Projections of Intense Tropical Cyclone Activity for the Late Twenty-First Century from Dynamical Downscaling of CMIP5/RCP4.5 Scenarios. *Journal of Climate*, 28(18), 7203–7224. <https://doi.org/10.1175/JCLI-D-15-0129.1>

Reply to the reviewers' comments

The following document provides a point-by-point reply to reviewers' comments. Comments are numbered, for example "R3C3" denotes the third comment from reviewer 3. Our replies follow each comment in blue-colored text. The replies explain how we performed the revisions, why, and where in the paper. All text quoted from the paper is marked with quotes (" "). In this document, when we report excerpts of text from the paper, text that has been revised or newly added is marked in blue, while deleted text is in purple strikethrough (where relevant).

Reviewer 1

Reviewer #1 (Remarks to the Author):

R1C1

Thanks for a timely and relevant contribution to the literature. The paper is well written and the Mexico case provides a good illustration, indeed. A big limitation is the **use of one single impact function**, irrespective of the nature of the exposure, neither whether direct (assets) or indirect impacts (revenue) are dealt with. This should be stressed more in the **Discussion(3) as few will read the Limitations section** (5, where this is revealed) - and one sentence should be spent on this in the abstract, too. The current mention of **"...smoothing modelling assumptions regarding e.g. the use of a common damage function for all assets" (page 13) does rather hide than reveal this.**

Reply: We thank the reviewer for the encouraging comments regarding the contribution of our manuscript to the literature, and about its clarity. We also thank the reviewer for the relevant comment on the damage function. We agree on the importance of clarifying in the paper why we opted for a single impact function, and the potential limitations and implications for our analysis that this modeling choice brings. In this regard, we have comprehensively addressed this comment in the paper's "Discussion" section and in the dedicated subsection, called "Limitations", in the "Methods" section. We preferred however not to discuss this limitation in the abstract, since the space limit would not allow us to clearly explain to the reader the related challenges and implications for the analysis, thus potentially generating confusion.

In the "Discussion", the paragraph about limitations now reads as follows:

"Our results are conditioned to the following limitations, some of which, however, apply more in general to the field of climate physical risk assessment.

We acknowledge that our analysis does not take into account all the possible sources of uncertainty. Thus, our results provide a quantification of losses conditional to the following specifications.

First, we rely on point-estimates of the characteristics of assets (e.g., residual life, monetary value, capacity) because confidence intervals on those estimates are not available.

Second, we take the point-estimate for Return Period (RP) and Expected Annual Impacts (EAI) of tropical cyclones (TC) impacts from the CLIMADA model under the standard setting (Aznar-Siguan and Bresch, 2019). Indeed, there is not yet an established way to calibrate the set-up that enables to attribute levels of uncertainty on the estimates of RP and EAI (Kropf et al., 2022). In addition, the data needed to properly calibrate the assumptions that would be required to estimate uncertainty is not available.

Third, to quantify the impact of tropical cyclones, we relied on traditional approaches in the literature (e.g. Bresch and Aznar-Siguan, 2021, Rana et al., 2022) that interpolate between current (2001-2020) and future (2081-2100) climate scenarios but they do not take into account the uncertainty on the evolution of the impacts.

Furthermore, we use the same damage function across all types of assets since data on damages by asset-type are not available. In this regard, publicly available datasets, such as EM-DAT (UCLouvain, 2009) only provide damages per event, at the country or (more rarely) at the subnational level.

Finally, we do not consider firms' adaptation efforts (such as assets' relocation, implementation of physical barriers, etc.) because this information is currently not available (Goldstein et al., 2019).

Further limitations are discussed in Methods.”

R1C2

As for prior art, not in the academic literature, but in essence carrying out the same analysis, you might consider to reference the following two reports:

S&P, 2015: The Heat Is On: How Climate Change Can Impact Sovereign Ratings, <https://www.spglobal.com/ratings/en/research/articles/151125-the-heat-is-on-how-climate-change-can-impact-sovereign-ratings-9425836>

S&P, 2015: Storm Alert: Natural Disasters Can Damage Sovereign Creditworthiness, <https://www.spglobal.com/ratings/en/research/articles/150910-storm-alert-natural-disasters-can-damage-sovereign-creditworthiness-9327571> (unfortunately both not freely available, but you can register at www.spglobal.com for free)

Reply: We thank the reviewer for the comment and for suggesting non-academic work. It is important to notice that there are significant differences - in terms of scope and methodology - between the work of our paper and the ones presented in the two studies by S&P mentioned by the reviewer. Most importantly, the methodology from S&P focuses on countries and their sovereign ratings, while we focus on corporates, and on their equity shares. In finance, the models and methodologies for risk assessment and pricing of sovereign bonds are very different from those applied to corporates and equity shares. The same applies in the context of climate financial risk assessment and risk pricing. For instance, country-level information from past disasters used by S&P for sovereign risk analysis is available from publicly available databases (e.g., EM-DAT)¹ while historical information at firm and asset level is missing. This is why we rely on a probabilistic risk

¹ <https://www.emdat.be/>

assessment to assess asset-level and firm-level damages. Moreover, and importantly, our methodology includes a dedicated treatment of the shock transmission channels and implications of climate physical risks on firms' business lines, economic sectors and on macroeconomic variables, captured with the use of a dedicated macroeconomic model (ICES).

We have now added to the manuscript a paragraph to clarify these differences between sovereign and corporate methodology in the "Methods", subsection "Climate physical risk assessment" as follows:

"Assessing physical risk on corporate securities differs significantly from assessing physical risk for sovereign ratings or debt, especially in terms of data availability. In fact, country-level information on past disasters is available from publicly available databases (e.g., UCLouvain, 2009), but historical firm-level information is missing. Thus, we rely on a probabilistic risk assessment approach to assess asset-level and firm-level losses. In addition, our methodology includes a dedicated treatment of the transmission channels and implications of climate physical risks on firms' business lines, on economic sectors and macroeconomic variables, the latter being captured with the use of a dedicated macroeconomic model (ICES)."

We have also linked this additional paragraph to our "Results", adding the following footnote in the subsection "Methodological framework":

"Moreover, data on past disasters at the asset and firm levels are missing. To overcome this problem, we use a damage function, see Methods. See Methods also for a discussion of the difference with the case of climate sovereign risk assessment."

While commercial methodologies are increasingly targeting climate risks, it is important to highlight that such methodologies are not publicly available and replicable, and this is a main limitation in the context of scientific and academic research, as well as of policy design. Furthermore, the literature has highlighted that commercial methodologies also suffer from very significant limitations, and diverge in their conclusions (Hain et al., 2022). To address this comment, we have added the following sentences to the "Introduction":

"Most often, the assessment of climate physical risks relies on commercial methodologies. However, these methodologies are proprietary, not fully transparent, not fully replicable, and often lead to diverging results (Hain et al., 2022). Moreover, commercial methodologies provide climate risk scores with different levels of aggregation (e.g. by firm or hazard, depending on the provider). While scores could be useful for some types of analyses (e.g., investigating the market premium for physical risk), they cannot be used as a proxy for the asset-level damages, which in turn are needed to inform climate financial valuation models and climate financial risk pricing."

Some further comments (as line numbering was missing, I've quoted the sentence my comment relates to):

R1C3

Page 2, 2nd last line: "...estimated in USD 250 billion per year,...", might better be phrased ...estimated to amount to...

Reply: We thank the reviewer for the suggestion. We have changed "estimated in" to "estimated to amount to" in the manuscript.

R1C4

Page 6, Figure 2, caption: "In the central regions of the country, assets are mostly not impacted (i.e. grey colour) as these regions are generally not interested by tropical cyclones.", you mean exposed, not interested, I presume. But what about torrential rain and flooding associated with tropical cyclones? In general, shorten the figure caption, as quite some text is redundant (e.g. "The location of the dot is determined by the latitude and longitude coordinates of the asset." three times).

Reply: we thank the reviewer for the comment. We have adjusted "interested by" to "exposed to" and generally shortened the caption removing redundant elements. The new caption reads as follows:

~~"Assets' distribution and direct impact of tropical cyclones on assets. All panels: assets are represented as dots. The position of the dots is determined by the latitude and longitude coordinates of the asset. The size of the dots is proportional to the standardized capacity of the asset (i.e. assets with larger capacity will have larger dots). Top-left panel: assets' geographical distribution. Assets are represented as dots, where the The colour of the dot describes the type of asset (e.g. mine, power plant, see legend in the bottom left of the panel) according to the legend in the bottom left of the panel. and the size of the dot is proportional to the standardized capacity of the asset (i.e. assets with larger capacity will have a larger dot). The location of the dot is determined by the latitude and longitude coordinates of the asset. Top-right panel: percentage of direct damages from tropical cyclones on assets under Expected Annual Impacts (EAI), RCP2.6, year 2050. The colour represents the percentage of direct damages as described by the colourbar to the right of the figure. As expected, all assets have a nearly zero direct impact under EAI all assets have a grey colour i.e. a nearly zero direct impact under EAI (i.e. grey colour). The size of the dot is proportional to the standardized capacity of the asset (i.e. assets with larger capacity will have a larger dot). The location of the dot is determined by the latitude and longitude coordinates of the asset. The red rectangle delimits the area which is zoomed in the bottom-left and bottom-right panels. Bottom-left panel: percentage of direct damages from tropical cyclones under a Return Period of 100 years (RP100), RCP2.6, year 2050. The colour represents the percentage of direct damages as described by the colourbar to the right of the figure. In some areas, for example the Jalisco region, direct damages are higher than the EAI case as shown by the yellow or orange colour. In the central regions of the country, assets are mostly not impacted (i.e. grey colour) as these regions are generally not interested by exposed to tropical cyclones. The size of the dot is proportional to the standardized capacity of the asset (i.e. assets with larger capacity will have a larger dot). The location of the dot is determined by the latitude and longitude coordinates of the asset.~~

Bottom-right panel: percentage of direct damages from tropical cyclones under a Return Period of 250 years (RP250), RCP4.5, year 2050. The colour represents the percentage of direct damages as described by the colourbar to the right of the figure. In some areas, for example the Jalisco region, direct damages are higher than the EAI case as shown by the red, orange, and yellow colour. In these areas, some assets are more impacted than under RCP2.6 (bottom-left panel), though differences are limited due to the short horizon (2050). In the central regions of the country, assets are mostly not impacted (i.e. grey colour) as these regions are generally not interested by exposed to tropical cyclones. (The size of the dot is proportional to the standardized capacity of the asset (i.e. assets with larger capacity will have a larger dot). The location of the dot is determined by the latitude and longitude coordinates of the asset. The colourbar to the right of the chart is common to the top-right, bottom-left and bottom-right panels and relates the colour of the dot to the percentage of direct damages from tropical cyclones.”

We transferred some information previously part of the caption to the discussion of the Figure. The amended parts now read as follows:

“However, losses from tropical cyclones are concentrated in certain areas of the country, such as the Jalisco region, as indicated by the yellow, orange, and red colours in the bottom panels of Figure 2. Other areas, like the central region, are not exposed to tropical cyclones as indicated by the grey colour. [...]

Moreover, for 13% of assets the losses conditioned to RP250 are 5 times larger than the losses conditioned to RP100. However, differences in losses across scenarios are limited due to the short time horizon and become more pronounced only toward the end of the century.”

Note that we have also rewritten the subsection to make a consistent usage of the terms “EAI”, “losses”, and “damages”. Definitions of the key notions used in the literature such as EAI, tail risk, Value-at-Risk, Return Period are now provided in the beginning of subsection 2 - Results - Methodological framework.

In addition, we have clarified in the “Methods”, “Climate physical risk assessment” subsection, “Asset-level assessment” subsubsection, the remark on torrential rain and flooding, using the following sentences:

“The damage function in Equation 1 considers only wind speed. This is a common assumption in the literature (Aznar-Siguan and Bresch, 2019, Dunz et al., 2021, Emanuel, 2011). Nevertheless, considering only wind speed limits the extent of the assessment of those hurricanes where rainfall and storm surges can account for high damage, despite the storm being less windy (Aznar-Siguan and Bresch, 2019).”

R1C5

Page 8, Figure 3: It is somewhat troubling to denominate acute shocks with positive numbers. Why did authors not denote these shocks as negative, as they did for chronic and combined shocks?

Reply: We thank the reviewer for this relevant comment. In the previous version we had assigned the acute shock a positive sign because it represented the variable η (Equation 9), i.e. the average asset-level damage scaled by residual life. $\eta+1$ is denoted as δ (Equation 9) and it enters the model to adjust the term g_L at the denominator (Equation 8). As such, η is a positive number for the purpose of calculations.

However, we agree that using different sign for the two shocks makes it harder to interpret the results in the chart. We have revised the sign now so that both shocks are shown using a negative sign. Thus, we have now changed Figure 3 to show $-\eta$. This represents the (scaled) damage to assets. We have updated the caption and the discussion of the figure accordingly. The new discussion of the figure reads as follows:

“Four main results emerge. First, chronic and acute shocks both contribute to higher equity shocks (see e.g. firm 1, mainly active in the fossil fuels sector). Second, firms in the same sector can have similar chronic shocks but very different acute shocks because their assets are in different locations (see e.g. firm 2 vs firm 3, active in the renewables sector). Third, firms may be affected by a similar acute shock but by different chronic shocks (see e.g. firm 3, renewables, vs firm 4, mining). Fourth, firms can have large acute shocks even if operating in different sectors (see e.g. again firms 3 and 4).”

Note that we have also rewritten the subsection to make a consistent usage of the terminology related to “Value at Risk”, “RP”, and “EAI”.

R1C6

Page 9, Figure 4: “Scatter plot for the joint EAI and RP250 equity loss distributions for the year 2040, 86 companies with available asset-level data”. Are asset values scaled to 2040? How? See also my comment iro Page 14.

Reply: We thank the reviewer for the comment and the opportunity to clarify this important point. Asset values are not scaled to 2040. We choose this approach for two reasons. On the one hand, our Climate Dividend Discount Model (CDDM) as described in Equations 7-8 is a reduced form model of the long-run growth (thus, assets, earnings, profits, dividends) of a firm. This reduced form approximation is well-grounded in the financial literature on discounted dividends and discounted cash flow models (see e.g., (Sharpe et al., 1999)). The idea of the reduced form model is to capture the average effect of climate change across all firm’s assets, for a long-run period that stretches beyond 2035. Thus, in the context of Equation 8, what matters is the relative value of damages (i.e. value of damages over value of assets that the firm bears), rather than the absolute value of the individual assets. As such, scaling asset values to 2040 is possible but does not have an impact on the results: since the damage function (Equation 1) returns damages as a fraction of value, increasing asset values would lead to increased absolute damages, but constant fraction of damages. On the other hand, this approach enables us to keep consistency with the modelling of climate shocks in the macroeconomic model ICES. In fact, ICES represents the impacts of climate change either as changes in productivity or in losses on physical capital and land. Thus, our focus on relative losses to assets is consistent with the treatment of physical capital in the macroeconomic model.

We have now added a paragraph to the “Methods”, subsection “Climate financial valuation: the Climate Dividend Discount Model”, subsubsection “Climate Dividend Discount Model”, to clarify this point:

“Importantly, in Equation 8, delta is computed using the relative damages to assets as calculated by the CLIMADA model. Thus, in our model the impact of acute shocks on the firm is captured in a reduced form as adjustment in the growth rate of the firm. In this treatment, the ratio of asset damages (as computed in Equation 1 using the damage function) links acute risks to the growth rate. Hence, it is not necessary to model the growth of assets explicitly. This approach is also consistent with the one followed in the macroeconomic model ICES. In fact, ICES represents the impacts of climate change either as changes in productivity or as losses on physical capital and land. Thus, the focus on relative losses to assets is consistent with the treatment of physical capital in the macroeconomic model.”

R1C7

Page 13: With respect to implications:

1) “...contributes to inform policymakers in the design and implementation of adaptation policies...” can you be more specific in that sense?

Reply: We thank the reviewer for the comment. We have now clarified the policies we refer to, including e.g. the European Climate Adaptation Strategy (<https://eur-lex.europa.eu/legal-content/EN/TXT/HTML/?uri=CELEX:52021DC0082>). The new paragraph reads as follows:

“Our analysis has important implications for several types of decision makers in climate finance. On the one hand, it contributes to inform policy makers in the design and implementation of adaptation policies, in particular in the current context of public budget constraints and rising public debt, both in the Global South and in the North.

For example, in the EU, the European Climate Adaptation Strategy called for *smarter adaptation* and highlighted the need to translate large volumes of climate information into customized and user-friendly tools (EC, 2021) . In this regard, our methodological framework sets a blueprint for the use of geolocalized climate data for climate finance analysis, and supports the Strategy’s objectives, such as closing the adaptation gap, integrating climate risks and resilience into fiscal frameworks, and scaling up finance for climate resilience.

In addition, our framework contributes to inform financial regulators and supervisory authorities to better assess the impact of climate physical risks for sovereigns, considering the composition of each country’s economy (e.g. contribution of firms and sectors to Gross Value Added and fiscal revenues), the impact on sovereign risk (e.g. via bonds’ yields and spreads) and the potential financial and regulatory response (e.g. monetary and prudential policies (ECB/ESRB 2022)).”

R1C8

2) “..supports financial institutions, such as insurance and reinsurance companies..”. This might be true for banks (see Westcott, M., Ward, J., Surminski, S., Sayers, P., Bresch, D.N.

and Claire, B., 2020. Be prepared: Exploring future climate-related risk for residential and commercial real estate portfolios. *The Journal of Alternative Investments*, 23(1), pp. 24-34. <https://jai.pm-research.com/content/early/2020/05/09/jai.2020.1.100.abstract>), but insurers and reinsurers in particular do apply natural catastrophe models to price their business and manage their capital already. As the latter use risk models with bespoke impact functions (much more detailed than as used in the present study), one would not expect them to benefit much from insights as presented here. But in a wider TCFD context, there are many private sector actors (real economy, not only financial sector) who would benefit a lot from such disclosure, not least investors... yet full transparency is needed, which means the models along the whole chain as presented in the paper need to be open-source and free-access.

Reply: We thank the reviewer for the comment. We acknowledge that there is indeed a different level of sophistication among investors. We have clarified this point and mentioned that insurers and reinsurers tend to already rely on sophisticated catastrophe models. We thus specified that the main beneficiaries in the financial sector are banks and investment/pension funds in the “Discussion”.

Furthermore, also in the “Discussion”, we highlighted the limits of portfolio diversification in the context of physical risks. Finally, we have noted that the methodology we propose is also relevant for non-financial actors who need to perform physical risk assessment and disclose physical risk metrics e.g. to meet requirements from the Corporate Sustainability Reporting Directive (CSRD)². We have highlighted the importance of this point considering the current limitations in firms’ preparedness in physical climate risk management and adaptation strategies (Goldstein et al., 2019). The new paragraphs read as follows:

“On the other hand, our methodology supports financial institutions in avoiding underestimations of exposures to physical risks.

In fact, while some financial actors such as insurers and reinsurers are relatively well-prepared in catastrophe risk management, others are less sophisticated (Westcott et al., 2020). This is for instance the case of banks’ lending to firms (which represents on average 40% of EU banks’ total assets)³ and of institutional investors (e.g. asset managers, pension funds and investment funds).

Importantly, even a well-diversified portfolio would bear high physical risks as extreme events are predicted to increase globally (IPCC, 2022). Thus, geographic and sectoral diversification have limited benefits for portfolio risk reduction. In contrast, our methodology enables investors to better identify physical risks at the asset and firm level, considering assets’ location. With this information, investment strategies, which may include a conditionality on adaptation investments from borrowing or invested firms can be implemented, improving banks’ and investors’ resilience to physical risks.

Finally, our analysis shows how firms could improve their climate risk management and facilitate physical risk disclosures under the Corporate Sustainability Reporting Directive

2

https://finance.ec.europa.eu/capital-markets-union-and-financial-markets/company-reporting-and-auditing/company-reporting/corporate-sustainability-reporting_en

³ [Risk Dashboard | European Banking Authority \(europa.eu\)](https://www.eba.europa.eu/en/risk-dashboard)

(CSRD, Directive (EU) 2022/2464). This is of key importance in light of the significant blind spots that still exist in firms' assessment of climate risks and in their adaptation strategies (Goldstein et al., 2019).”

Regarding transparency, we believe the manuscript contains the information necessary to replicate the study by third parties. Unfortunately, due to the fact that underlying data have been sourced from commercial providers, it is not possible for us to make the underlying or derived data publicly available. We have clarified this in the (newly added) “Data availability statement”, which reads as follows:

“The data that support the findings of this study are available from S&P and Refinitiv Eikon, but restrictions apply to the shareability of these data, which were used under license for the current study, and so are not publicly available. Data are however available from the authors upon reasonable request and with permission of S&P and Refinitiv Eikon.”

At this stage, we can make parts of the code available upon reasonable request and the agreement of non-commercial use, as clarified in the (newly added) “Code availability statement” (required by the journal), which reads as follows:

“The codes that support the findings of this study can be made available upon request for the purpose of academic research.”

In particular, requests should be accompanied by an agreement to not use the code for commercial purposes.

R1C9

Page 14: With respect to asset data, how are future projections handled? I did not find a description of the scaling of assets (and related turnover or revenues) with SSPs (they are mentioned further below in the context of the macroeconomic assessment).

Reply: We thank the reviewer for the comment and the opportunity to clarify this point. As discussed in our reply to R1C6, we do not scale the value of assets as the variable that matters for the adjustment of the growth rate in Equation 8 is the relative damage to assets. This remains constant when increasing asset values, following Equation 1. Moreover, this enables us to keep consistency with the macroeconomic model where climate shocks are translated into damages to capital stock. Asset values are also not scaled to SSPs, following the same reasoning. We have added this explanation in the “Methods”, subsection “Climate financial valuation: the Climate Dividend Discount Model”, subsubsection “Climate Dividend Discount Model”, as discussed in our reply to R1C6.

R1C10

Page 15: “The grid is set at 0.2 degrees of latitude/longitude, for a total of

14,076 centroids matched to hazards.” There would be pre-computed probabilistic tropical cyclone hazard sets (today and RCPs) at a resolution of 150 arcsec (4 km) available via the CLIMADA data API (<https://climada.ethz.ch/data-types/>). Why did the authors resort to a coarser resolution of 0.2 degrees?

Reply: We thank the reviewer for this very relevant remark. We did not use the API for several reasons. First, we built on (Aznar-Siguan and Bresch, 2019) for the probabilistic risk assessment. This implies the usage of 50 synthetic tracks per real track; and the usage of a large enough sample of historical data, starting in 1950. This approach does not allow for using the API. Moreover, API simulations are available only for 10 synthetic tracks to the best of our testing⁴, and for historical years 1980-2020. Thus, the API data offer less tracks for a shorter historical period, which would be inconsistent with the literature we are building on. Second, the format of the API data is inconsistent with the model architecture that we have implemented. In fact, the API provides data at reference years 2040, 2060, 2080. This is inconsistent with the 5-year steps that are necessary to make the probabilistic risk assessment comparable with the macroeconomic assessment of chronic risks (2035-2040-2045-2050). Thus, it was not possible to use the API directly.

We used the coarser resolution as a compromise between accuracy and computational cost. It is also important to notice that, for some assets, the polygon delimiting the area of the asset is larger than 150 arcsec (see e.g. Maus et al., 2020 for mines), or the location is not classified as “exact” from S&P or Refinitiv Eikon. As such, the coarser resolution enables us to account for some of the spatial uncertainty in the asset distribution.

Nevertheless, to address the reviewer’s comment, we tested the API at the asset-level to investigate differences in damages. To do so, we considered the year 2040, that is, the only year overlapping between the API’s computed datasets and our computations. We tested damages using EAI, RP100 and RP250 for RCPs 2.6, 4.5, 6.0. The only remaining difference between the API set-up and our set-up, outside of grid resolution, is the time-step for wind speed interpolation (0.5 hours in our simulation, 1 hour for the API). We have added the results of this comparison to the Supplementary Materials, in subsection “Asset-level damages with a finer grid”, as described below. In summary, the differences between the finer and coarser grids are negligible in the case of EAI. In the case of RP250, the average deviation is 0.38% (i.e., the damages are higher on average for the finer grid). The 75th percentile of changes in damages when moving from the coarser to the finer grid is 0.2%. The maximum is 14%. Only 0.66% of assets experience an increase in damages larger than 10%; and 1.99% of assets have an increase in damages larger than 5%. Thus, we can conclude that asset-level damages would increase for some assets using the finer grid. However, at the firm level, the conclusion is not straightforward. While we can expect an increase in general, some netting may occur as the minimum deviation is -4%, i.e. there are

⁴ We acknowledge that the CLIMADA documentation states that simulations on 50 tracks are available, see e.g. here:

https://climada-python.readthedocs.io/en/latest/tutorial/climada_util_api_client.html. Unfortunately, extensive testing showed that this is not the case on our end. We tested the API using two machines and three different versions of CLIMADA (one machine running v.3.3.2(dev) and v.3.1.1.dev0; the other running 3.2.0) and in all cases we could only access the 10 tracks version.

assets for which damages are higher with the coarser grid. We have discussed these issues in the section “Asset-level damages with a finer grid” in the Supplementary Materials:

“Asset-level damages with a finer grid”

“To complement our analysis, we investigate the effect of using a finer spatial grid (150 arcsec, or approximately 4.5 km at the Equator, instead of 0.2 degrees, or approximately 22 km at the Equator) on the asset-level damages. To do so, we use CLIMADA’s data API⁵ to source pre-computed, 150 arcsec gridded tropical cyclones maximum intensities over Mexico. The API data are simulated using 10 synthetic tracks for each real track, historical data for years 1980-2020, reference year for climate conditions 2040, and a wind speed interpolation step of one hour. Hence, there are four key differences between our set-up and the one used in the API: the grid spacing, the number of simulated tracks, the interpolation step, and the reference period for historical data. We use 2040 as a reference year for the comparison since it is the only common year between our set of years (2035, 2040, 2045, 2050) and the API’s set of years (2040, 2060, 2080). For both the API data and our original assessment (see Methods), we compute asset-level damages as a percentage of asset value.

We compare the two cases and find that the difference between the finer and coarser grid is negligible (<0.007%) in the case of EAI. When considering RP250, we find a small effect in the adjustment of the damages (slight increase or decrease). The maximum of the average difference in damages across all assets using the finer grid with respect to the coarser grid is 0.38% (i.e. 0.44% with coarser grid vs 0.82% with finer grid). The average difference does not vary significantly by asset type, ranging from a minimum of 0.07% for power plants to a maximum of 1.09% for liquefaction plants.

Importantly, for the majority of assets, differences in damages using the finer and coarser grid are negligible, with the 75th percentile of the distribution of differences being 0.2%. In a few cases, some significant changes occur, with maximum higher damage from the finer grid being 14% and minimum lower damage from the finer grid being -4%. However, only 0.66% of assets have a higher damage from the finer grid that exceeds 10% in the RP250 case. Thus, we can conclude that using the finer grid leads to some assets suffering more damages in our framework. At the firm level, results are less clear, as some netting may occur (i.e. some firm’s assets may be more damaged, but others may be less damaged, with the two effects reducing firm-level changes). Still, on average, firm- and portfolio-level damages can be expected to marginally increase. Importantly, we tested the refined grid also in the proxyl case finding that no change in damages occurred, neither for EAI nor for RP250. Thus, our main results on underestimation are not impacted by the usage of a coarser grid.

Full statistics for the comparison are reported in Supplementary Table III below.

⁵ https://climada-python.readthedocs.io/en/latest/tutorial/climada_util_api_client.html

Statistic	4026_eai	4045_eai	4060_eai	4026_rp250	4045_rp250	4060_rp250
mean	0.00	0.01	0.01	0.23	0.38	0.34
std	0.02	0.02	0.02	1.21	1.52	1.37
min	-0.04	-0.05	-0.04	-4.14	-3.99	-3.57
25%	0.00	0.00	0.00	0.00	0.00	0.00
50%	0.00	0.00	0.00	0.01	0.02	0.01
75%	0.00	0.00	0.00	0.15	0.22	0.19
max	0.19	0.24	0.21	11.48	14.38	12.80

Table III: Summary statistics for the difference in asset-level damages obtained comparing the 0.2 degrees grid used in this study with the 150 arcsec grid sourced from CLIMADA's API. Four key differences exist between our set-up and the one used in the API: the grid spacing, the number of simulated tracks, the interpolation step, and the historical data timeframe. Summary statistics are presented by row, in the following order: mean (*mean*), standard deviation (*std*), minimum (*min*), 25th percentile (25%), 50th percentile (50%), 75th percentile (75%), and maximum (*max*). Each column represents a measure of damages (EAI or RP250) for a different scenario, for the reference year 2040, in the following order: EAI, year 2040, RCP2.6 (4026_eai); EAI, year 2040, RCP4.5 (4045_eai); EAI, year 2040, RCP6.0 (4060_eai); RP250, year 2040, RCP2.6 (4026_rp250), RP250, year 2040, RCP4.5 (4045_rp250); RP250, year 2040, RCP6.0 (4060_rp250). Values in the cells are expressed in percentages. Source: authors."

We have referenced this supplementary section in the "Methods" ("Climate physical risk assessment" subsection, "Asset-level assessment" subsubsection), using the following note where we discuss the grid spacing:

"We also tested the effect of using a finer grid on asset-level damages for the year 2040, see Supplementary section "Asset-level damages with a finer grid". The comparison shows that using a finer grid could lead to slightly higher asset-level damages."

R1C11

Page 16: "We follow the calibration provided by (Dunz et al., 2021) for Mexico." Would be worth comparing with the parameters as determined by Eberenz, S., Lüthi, S., and Bresch, D. N., 2021: Regional tropical cyclone impact functions for globally consistent risk assessments, Nat. Hazards Earth Syst. Sci., 21, 393-415, <https://doi.org/10.5194/nhess-21-393-2021>.

Reply: We thank the reviewer for the comment. We have added the comparison between the calibrations from (Dunz et al., 2021) and (Eberenz et al., 2021) in section "Methods", subsection "Climate physical risk assessment", subsubsection "Asset-level assessment" of the manuscript, using the following sentences:

"Other calibrations exist in the literature, for example (Eberenz et al., 2021), which sets $\$W_{\{thresh\}}=92.52\text{km/h}$ and calibrates $\$W_{\{half\}}$ with two different approaches to either 214.56km/h or 238.68km/h. In comparison, the calibration by (Dunz et al., 2021) may

overestimate the damages from low-category hurricanes and underestimate the damages from high-category hurricanes. Both the calibrations by (Dunz et al., 2021) and (Eberenz et al., 2021) are based on the shape of the damage function proposed by (Emanuel, 2011). However, the former calibration is performed on disaster damage data from Mexico only, while the latter is performed on disaster damage data from Mexico and the Caribbean (for W_{half}), and on disaster damage data from the US (for W_{thresh}), consistently with (Emanuel, 2011). We use the calibration by (Dunz et al., 2021) for this study, as it is specific to Mexico only.”

R1C12

Page 16: “Importantly, we keep the damage function constant across asset types.”. This is a rather crude assumption, as authors state “Calibration of asset-level damage functions is left for further research”. It would be worth checking for stability of the results using variants of impact functions. The calibration as stated is based on damage to general property, while industrial assets usually show lower a loss burden. Business interruption (or revenue loss) is yet another story. Hence a proper sensitivity analysis might be considered, see Kropf, C., Ciullo, A., Otth, L., Meiler, S., Rana, A., Schmid, E., McCaughey, J. W., and Bresch, D. N., 2022: Uncertainty and sensitivity analysis for probabilistic weather and climate-risk modelling: an implementation in CLIMADA v.3.1.0. *Geosci. Model Dev.*, 15, 7177-7201, <https://doi.org/10.5194/gmd-15-7177-2022>

Reply:

We thank the reviewer for this comment. We agree with the reviewer on the importance of clarifying why we opted for a single damage function and the potential limitations and implications for our analysis. This limitation stems from the lack of data on disaster damages by asset-type. In fact, publicly available disaster datasets, such as EM-DAT⁶, provide only aggregated damages per event, at the country or subnational level.

Thus, it is not possible to calibrate asset-specific damage functions and reverting to a single one is a standard approach in the literature (see e.g. Dunz et al., 2021, Bresch and Aznar-Siguan, 2021). We have acknowledged and discussed this limitation both in the “Discussion” and the “Methods” (subsection “Limitations”) sections. We have also included this point in our answer to R1C1.

We also thank the reviewer for pointing us toward the uncertainty module introduced in (Kropf et al., 2022). While we fully agree on the importance of sensitivity analysis and uncertainty analysis, we believe the analysis would be severely limited in lack of proper calibration. In fact, as noted also in (Kropf et al., 2022) “Defining the appropriate input variable uncertainty and identifying the relevant input parameters for a given case study are not trivial tasks”; and “not all uncertainties can be described with the shown method [...]; only the uncertainty of those input parameters that are varied can be quantified, and even for these input parameters, defining the probability distribution is subject to strong uncertainties, often being based only on educated guesses.” Moreover, the authors note that “the choice of

⁶ <https://www.emdat.be/>

probability distribution can have a strong impact on the resulting model output distribution and sensitivity”. Thus, while it would be technically possible to add an uncertainty analysis of the shape and thresholds of the damage function, it would be of limited scientific value from our perspective due to the lack of data for the calibration of the uncertainty distribution.

R1C13

Page 22: “To perform the firm-level analysis, we replace asset-level data with one location per company and use information on physical risk at this location to proxy physical risks for the company.” Yet you stated on page 3 “...previous works on risk assessment have resorted to approximating the location of economic activities of each firm with its headquarter location”. Could you please explain why you deem a single point a good proxy for the asset portfolio (distributed across Mexico - and beyond) for a company?

Reply: We thank the reviewer for this comment that allows us to clarify our message. From the comment, it appears there has been a misunderstanding in the purpose of this section. We do not claim that a “single point” (i.e., firm-level estimate, now renamed as “proxy data”) is a good proxy of the asset-level assessment of physical risk. On the contrary, our goal is to show the discrepancy between the two types of estimates and thus provide evidence for the importance of carrying out the asset-level analysis, which is the core of our manuscript. To clarify our terminology, we have now referred to the case where only one location per firm is used as “proxy data”, as opposed to “firm-level” data.

In essence, our workflow proceeds as follows. We first introduce our methodological framework (Figure 1), which is based on asset-level information. This represents data on the production facilities owned by firms. Then, we illustrate the different steps of the methodology in an application to Mexico, at three different levels i.e.: (i) at the asset level (subsection “Acute risk at the asset level”), (ii) at the firm level (subsections “Acute vs chronic risks at the firm level” and “Tail acute risks lead to large economics and financial losses for firms”), and (iii) at the portfolio level (subsection “Substantial underestimation of investors’ losses from neglecting tail acute risks”). All these results are obtained using in the computation the complete set of asset-level information available. Lastly, we are interested in understanding what happens when asset-level information is not available or it is neglected. To perform this additional analysis, we introduce the notion of “proxy data” analysis, which indicates the computation of the damages as if the location of all the firms’ assets were the headquarter (HQ) of the firm. This counterfactual analysis enables us to quantify the underestimation of risk associated with the approach of neglecting asset-level information (subsection “Neglecting asset-level information leads to a relevant underestimation of losses for investors”). The methodology underpinning the counterfactual analysis is discussed more in detail in the “Methods” (subsection “Firm-level analysis”, now renamed to “Analysis using proxy data”).

We present this counterfactual analysis in our study to highlight the importance of using asset-level data. In fact, in light of the poor availability and quality of asset-level data, academics and practitioners have resorted to using proxy locations for physical risk (see e.g. Hong et al, 2019). Still, the potential mistakes and underestimation of risk in doing so have

not been yet highlighted and quantified. This is why we believe this is a relevant contribution of our paper.

We acknowledge that the presentation of these aspects could be improved and we are grateful to the reviewer for having pointed this out. To clarify this point in the manuscript, we now refer to the analysis conducted using only one location as “proxy data” analysis. Furthermore, we have revised the sentence flagged by the reviewer and the title of that subsection to highlight that this is a counterfactual analysis. The new title of the subsection is: “Analysis using proxy data” and the first sentences of that subsection read:

“We compare the portfolio-level results for losses computed using proxy data vs asset-level data. The purpose of this analysis is to quantify the relevance of the underestimation of physical risk that stems from neglecting asset-level data. To compute the results with proxy data, [...]”

Lastly, in the subsection “Substantial underestimation of investors’ losses from neglecting tail acute risks” where the proxy data concept is first introduced, we have now specified its meaning. The new sentence reads as follows: “In absence of granular asset-level data, a possible approximation, which we refer to as use of “proxy data”, consists of replacing the set of locations where the firm operates by only one location, generally the country, city or street of its headquarter (see e.g., (Hong et al., 2019)).”

References cited

(Aznar-Siguan and Bresch, 2019) Aznar-Siguan, G., & Bresch, D. N. (2019). *CLIMADA v1: a global weather and climate risk assessment platform*. *Geoscientific Model Development*, 12(7), 3085–3097.

(Bresch and Aznar-Siguan, 2021) Bresch, D. N., & Aznar-Siguan, G. (2021). *Climada v1. 4.1: Towards a globally consistent adaptation options appraisal tool*. *Geoscientific Model Development*, 14 (1), 351–363.

(CSRD, Directive (EU) 2022/2464) European Parliament and Council of the European Union. (2022). *Directive (eu) 2022/2464 of the european parliament and of the council of 14 december 2022*.

(Dunz et al., 2021) Dunz, N., Hrast Essenfelder, A., Mazzocchetti, A., Monasterolo, I., & Raberto, M. (2021). *Compounding COVID-19 and climate risks: the interplay of banks’ lending and government’s policy in the shock recovery*. *Journal of Banking & Finance*, 106306.

(Eberenz et al., 2021) Eberenz, S., Lüthi, S., & Bresch, D. N. (2021). *Regional tropical cyclone impact functions for globally consistent risk assessments*. *Natural Hazards and Earth System Sciences*, 21, 393–415.

(EC, 2021) European Commission (EC). (2021). *Communication from the commission com(2021) 82 final. forging a climate-resilient europe - the new eu strategy on adaptation to climate change*.

(ECB/ESRB 2022) ECB/ESRB. (2022). *The macroprudential challenge of climate change* (tech. rep.). ECB/ESRB Project Team on climate risk monitoring.

(EM-DAT) UCLouvain, C. I. (2009). *EM-DAT: the CRED/OFDA international disaster database*.

(Emanuel, 2011) Emanuel, K., & Jagger, T. (2010). *On estimating hurricane return periods*. *Journal of Applied Meteorology and Climatology*, 49 (5), 837–844.

(Goldstein et al., 2019) Goldstein, A., Turner, W. R., Gladstone, J., & Hole, D. G. (2019). *The private sector's climate change risk and adaptation blind spots*. *Nature Climate Change*, 9 (1), 18–25.

(Hain et al., 2022) Hain, L. I., Kölbl, J. F., & Leippold, M. (2021). *Let's get physical: comparing metrics of physical climate risk*. *Finance Research Letters*, 102406.

(IPCC, 2022) IPCC. (2022). Summary for policymakers. In H.-O. Portner, D. Roberts, M. Tignor, E. Poloczanska, K. Mintenbeck, A. Alegria, M. Craig, S. Langsdorf, S. Loschke, V. Moller, A. Okem, & B. Rama (Eds.), *Climate change 2022: impacts, adaptation and vulnerability. Contribution of Working Group II to the Sixth Assessment Report of the Intergovernmental Panel on Climate Change* (pp. 3–33). Cambridge University Press.

(Kropf et al., 2022) Kropf, C. M., Ciullo, A., Otth, L., Meiler, S., Rana, A., Schmid, E., McCaughey, J. W., & Bresch, D. N. (2022). *Uncertainty and sensitivity analysis for probabilistic weather and climate-risk modelling: an implementation in CLIMADA v.3.1.0*. *Geoscientific Model Development*, 15 (18), 7177–7201.

(Maus et al., 2020) Maus, V., Giljum, S., Gutschlhofer, J., da Silva, D. M., Probst, M., Gass, S. L., ... & McCallum, I. (2020). *A global-scale data set of mining areas*. *Scientific data*, 7(1), 289.

(Rana et al., 2022) Rana, A., Zhu, Q., Detken, A., Whalley, K., & Castet, C. (2022). *Strengthening climate resilient development and transformation in viet nam*. *Climatic Change*, 170 (1-2), 4.

(Sharpe et al., 1999) Sharpe, W. F., Alexander, G. J., & Bailey, J. V. (1999). *Investments* (6th). Prentice-Hall.

(Westcott et al., 2020) Westcott, M., Ward, J., Surminski, S., Sayers, P., Bresch, D. N., & Claire, B. (2020). *Be prepared: Exploring future climate-related risk for residential and commercial real estate portfolios*. *The Journal of Alternative Investments*.

Reviewer 2

Reviewer #2 (Remarks to the Author):

The authors clearly put in a lot of effort to integrate several databases in order to be able to assess financial losses when neglecting asset-level information and acute risks. I have several issues with the study though. Most important:

We thank the reviewer for the useful comments that we addressed below.

R2C1

1. The authors make the claim that neglecting asset-information and acute risks results in an underestimate of investor losses due to climate risks. I do not find this claim particularly interesting as it is true by definition. More interesting would be the question whether there is any evidence that investors (or policy makers or insurance companies) are actually underestimating these risks. Unfortunately the study is silent on this.

Reply:

We thank the reviewer for the comment, that helps us clarify the aims and contribution of the paper. We believe there might have been a misunderstanding which we are happy to try and clarify.

Let us focus first on the comparison between assessment of risk using asset-level information as opposed to the assessment using approximated location, which we refer now in the paper to as using “proxy data”.

To be precise, it is not correct to say that neglecting asset-information leads by definition to underestimation of risk. In fact, the relation between the two assessment depends on where firms’ assets and headquarter (HQ) are located. In principle, the mis-estimation can go both ways: there could be cases where assets are more exposed to physical risks than HQ, and cases where the opposite is true. For example, a firm could be headquartered in the Bahamas, but have assets only in Central Europe. For this particular firm, risks from tropical cyclones would be very high if computed based on headquarters in the Bahamas but near zero once accounting for the assets in Central Europe. Similarly, a firm may be headquartered in a coastal city in Florida, such as Miami, and have assets either in Central Florida or in other US states where tropical cyclone risk is much lower or non-existent. Multiple examples can be provided for this latter case.

Let’s take the example of Forza X1, Inc., which is a manufacturer of electric engines for boats. The firm is headquartered in Fort Pierce, Florida, is listed on the NASDAQ (ticker: FRZA), and has been invested by, amongst others, Wells Fargo, Morgan Stanley, Vanguard, and UBS.⁷ The firm owns a testing facility in Fort Pierce, Florida, and is currently building a 10.5 million USD production facility in McDowell County, North Carolina.⁸ While both Florida and North Carolina are exposed to hurricanes, the risk in Fort Pierce is much higher than the risk in McDowell County. Hence, for this firm, measuring hurricane risk using the HQ (Fort

⁷ Source: Refinitiv Eikon.

⁸ Source: <https://ir.forzax1.com/financial-information/financial-results>

Pierce, Florida) results in an overestimation of risk with respect to assets (McDowell County, North Carolina).

The firms Nextera Energy Partners, LP (listed on the NYSE, ticker: NEP) and Aura Minerals Inc, (listed on TSE, ticker: ORA) are in a similar situation. Nextera Energy Partners, LP is headquartered in Juno Beach, Florida and has a portfolio of electricity-generating assets across several US states, with 26% of capacity located in high-risk hurricane states.⁹ In contrast, Aura Minerals Inc is a mining firm, headquartered in Miami, with several mines across South America, of which only one is in a high-risk country for cyclones (namely, Honduras).¹⁰ In both cases, the risk measured at HQ is likely higher than the risk measured taking into account the ensemble of the assets, since the HQ is fully exposed to hurricanes but only a part of the assets are.

Thus, it is not true by definition that the risk computed using assets is higher than the risk computed using proxy data.

In anycase, the magnitude of the possible underestimation of risk emerging from neglecting asset-level data has not been documented in the literature yet. This is relevant because it could give rise to mispricing in financial markets.

Moving now to acute risk: we agree with the reviewer that considering acute and chronic risks leads to higher losses than considering chronic risk only. This is true by definition.

However, our paper aims to make two contributions in this regard: 1) we document the magnitude of the underestimation, which was unknown so far, in a relevant case study; 2) we highlight the importance of considering tail acute risks, using return periods, for climate financial risk assessment.

Lastly, we agree with the reviewer that the question of whether investors underestimate climate physical risks in their portfolios or not is an interesting one. However, a benchmark methodology to measure physical risk is necessary in the first place in order to answer this question. In fact, a transparent and comprehensive assessment of physical risks is a precondition to investigate the existence of a portfolio-level underestimation of risks. Moreover, additional data, for example questionnaires, would be needed to understand whether investors underestimate physical risks or not in practice.

In our study, we focus on the methodological aspects of the underestimation, and show that investors need to extend their focus to include asset-level data and tail acute risks.

In order to address the reviewer's concern, we have clarified in the "Introduction" that this study quantifies the magnitude of risk underestimation and that the methodology can subsequently be used to strengthen internal risk assessment and risk management practices, with the following sentences:

"Still, the magnitude of the resulting misestimation of losses using such proxy data has been unknown so far. Here, we show that, in our sample, the approximation leads to an

⁹ Source: authors' elaboration on S&P data and list of high-risk hurricane states from <https://www.statista.com/statistics/1269483/number-of-hurricanes-that-made-landfall-in-the-us-state/>

¹⁰ <https://auraminerals.com/en/operations/>

underestimation of climate-related losses of up to 70%, in terms of relative difference on Value at Risk (VaR), compared to the computation based on granular asset-level/plant-level data.”

[...]

“By quantifying these potential underestimations, our methodology enables investors and corporations to better integrate climate physical risks in their internal risk assessment and risk management processes.”

R2C2

2. The authors use granular plant level data to construct firm level exposure to climate risk and claim to be the first to do this. They focus on Mexican firms and in particular risks related to cyclone damage. The authors do not seem to be aware of Moody's 427 measure of climate stress exposure for (more economically relevant) S&P500. This measure, similar to this paper, is based on the locations of physical assets such as production plants and offices which is then aggregated to the company level. Moody's 427 provides risk scores in several risk categories: heat stress, water stress (drought), flood (corporations only), extreme rainfall (municipalities only), hurricane, and sea level rise. While the authors of this study focus on cyclone risk, Acharya et al. (2022) convincingly show that investors are concerned with heat stress but do not seem overly concerned with the other risk categories (including cyclone risk).

Acharya, V. V., Johnson, T., Sundaresan, S., & Tomunen, T. (2022). Is physical climate risk priced? Evidence from regional variation in exposure to heat stress (No. w30445). National Bureau of Economic Research.

Reply: We thank the reviewer for the comment. We are aware of Moody's 427 scores and indeed our work aims to improve such state of the art private solutions to climate risk assessment. The reason is that such currently available solutions, including Moody's 427, are neither transparent about how firm and asset-level damages and final scores are calculated, nor about the assumptions used, and thus cannot be tailored to specific analyses. Thus, they are of little use for scientific analyses. We have acknowledged the fact that commercial solutions for physical risks exist in the market in the “Introduction”, with the following sentence:

“Most often, the assessment of climate physical risks relies on commercial methodologies. However, these methodologies are proprietary, not fully transparent, not fully replicable, and often lead to diverging results (Hain et al., 2022). Moreover, commercial methodologies provide climate risk scores with different levels of aggregation (e.g. by firm or hazard, depending on the provider). While scores could be useful for some types of analyses (e.g., investigating the market premium for physical risk), they cannot be used as a proxy for the asset-level damages, which in turn are needed to inform climate financial valuation models and climate financial risk pricing.”

Our manuscript contributes to fill this gap providing a transparent, replicable and tailorable, science-based methodology. In fact, commercial methodologies suffer from important limitations and, as a result, diverge significantly in terms of conclusions (Hain et al., 2022). For instance, Moody's 427's approach suffers from significant limitations. Moody's 427 data are provided to users as scores, which are relative in nature and thus do not provide information about the asset-level damages that a firm could suffer. Moreover, the data and methodologies with which these scores are generated remain a black-box for both researchers and investors. Finally, the scores are limited to certain types of assets only, with an important geographical skew. Thus, Moody's 427 data are of limited use to us as our purpose is to provide the full pipeline for physical risk analysis. Finally, the fact that the methodology from Moody's 427 is proprietary makes it difficult to transparently assess its limitations, differently from ours. While the scores from Moody's 427 are provided, according to the reviewer, for the S&P500, it is important to notice that the index includes many financial firms. However, computing physical risk scores for a financial firm using asset-level data is extremely complex, as it requires very granular information on portfolio composition that is generally not available. This raises further questions about the accuracy of the scores for financial firms in the S&P500, that unfortunately cannot be clarified due to the proprietary methodology.

We thank the reviewer for flagging the paper from Acharya and coauthors (Acharya et al., 2022). We have reviewed the paper and believe its conclusions do not invalidate the relevance of our study for two reasons. First, (Acharya et al., 2022) focus on US securities and on the US market, thus the conclusions may not generalize to other markets. In addition, (Acharya et al., 2022) use 427's scores of physical risk which suffer from the significant limitations that we discussed above. Moreover, in the context of tropical cyclones, it is important to notice that US impacts are not a good proxy for impacts in other regions (see e.g. (Eberenz et. al, 2021)), and important differences exist in calibrated impact functions from tropical cyclones between US and other regions.

Second, other studies in the literature show different results than (Acharya et al., 2022) using the same 427's scores. For example, (Gostlow 2022) shows that hurricanes command a positive risk premium. Furthermore, as noted also by (Acharya et al., 2022), "Correa et al. (2021) find evidence that changes in hurricane risk exposure affect corporate loan spreads". Hence, the evidence is still not conclusive with respect to the pricing of physical risks, one of the reasons being the methodologies used, thus justifying the need for physical risk assessment methodologies such as the one we introduce to enable further analysis.

Finally, the reviewer points out the higher financial relevance of the US with respect to Mexico in the context of physical risk pricing. Importantly, while we present an application to Mexico, our methodology is applicable to all countries. In fact, with available data, the methodology can be replicated for other countries, hazards, and securities. We have specified this in the "Discussion", with the following sentences:

"Our methodology can be applied beyond the Mexico case presented in this study. In fact, the presented steps can be tailored to assess physical risks for different countries (e.g. the US), hazards (e.g. floods) and securities (e.g. bonds, after tailoring the financial model). The main limitation to further applicability is represented by data availability."

References cited

- (Acharya et al., 2022) Acharya, V. V., Johnson, T., Sundaresan, S., & Tomunen, T. (2022). *Is physical climate risk priced? Evidence from regional variation in exposure to heat stress* (No. w30445). National Bureau of Economic Research.
- (Correa et al., 2021) Correa, R., A. He, C. Herpfer, and U. Lel (2021). *The rising tide lifts some interest rates: Climate change, natural disasters and loan pricing*. Federal Reserve Board Working Paper.
- (Eberenz et. al, 2021) Eberenz, S., Lüthi, S., & Bresch, D. N. (2021). *Regional tropical cyclone impact functions for globally consistent risk assessments*. *Natural Hazards and Earth System Sciences*, 21, 393–415.
- (Gostlow 2022) Gostlow, G. (2021). *Pricing Physical Climate Risk in the Cross-Section of Returns*. Available at SSRN 3501013.
- (Hain et al., 2022) Hain, L. I., Kölbel, J. F., & Leippold, M. (2021). *Let's get physical: comparing metrics of physical climate risk*. *Finance Research Letters*, 102406.

Reviewer 3

Reviewer #3 (Remarks to the Author):

This article seeks to provide a methodology for the quantification of physical climate risks and a procedure for translating these risks to financial losses at the company and portfolio level.

The article illustrates this methodology, and the procedures involved, through a sample of listed firms and activities located in Mexico, by assessing their exposure to (and the financial consequences of) tropical cyclones (TCs). It does so in five steps. First, through the construction of an asset-level database that connects financial information at the level of a firms' business lines to physical asset exposures. Second, by using the CLIMADA model to perform probabilistic assessment of damages from TCs at each asset location for different Representative Concentration Pathways (RCP2.6, 4.5, 6.0) at different time periods (2035, 2040, 2045, 2050). Third, by connecting these acute impacts with sector-level chronic impacts, as computed using the ICES model. Fourth, by developing a Climate Dividend Discount Model (CDDM) that connects these asset-level impacts on a company, to provide a climate-adjusted stock value. Fifth, by translating these impacts to the investor through the calculation of portfolio Value at Risk (VaR).

Through the above, the authors assert investor losses are underestimated by up to 70%, when asset-level information is neglected, and by up to 82%, when acute risks are neglected.

The problems the authors are trying to solve for, notably the assessment of asset-scale physical climate risk at the portfolio-level, is one that is critical to financial system actors and one that the investor community is grappling with. More broadly, the authors refer to challenges in assessing the impact of climate physical risks at the scale of an asset, arising from the computing limitations of current generation climate models and model uncertainty. The authors are to be commended for their work in attempting to solve the problem of

portfolio-level risk assessment and the challenges of incorporating asset-level data.

My chief concerns arise with regard to these challenges, as well as from an understanding of what constitutes climate risk. In the main, these concerns the first and second parts of the five-step framework developed. It is to these that I will speak to as this is where I am more qualified.

Asset-level risk (for extremes)

R3C0

The approach to incorporating asset-level data into portfolio-level analysis is to my understanding novel and to be commended. The authors have, with some stated limitations (including incomplete data and estimations based on these), built a database of physical asset information, including location, production capacity, monetary value, useful residual life, technology, operating status and ownership. In addition, the authors have linked this data to probabilistic assessments of future acute (hurricane) and chronic risks, to assess the exposure of these assets.

Reply: We are very grateful to the reviewer for the encouraging comment that recognizes the relevance and scope of our work.

R3C1

For the purpose of climate risk assessment, in particular risk assessment for extremes, risk is comprised of three elements: hazard, exposure and vulnerability (Cardona et al., 2012). While the hazard (hurricanes) and exposure (assets located in the path of the hurricane) are incorporated into the methodology developed, vulnerability (engineering of individual assets to particular standards, elevation, land surface type, tidal behaviour etc.) has to my understanding not. The relative vulnerability of a particular asset can, however, be instrumental in determining its preparedness for a particular hazard and, in this instance, its ability to adapt to hurricanes of greater intensity and frequency. Some assets may, for example, already be regularly exposed to TC risk and will accordingly be located at an elevation, on a land surface type and to a standard that renders them less vulnerable than others. I do not, for one instance, suggest that this problem is easy to overcome but would suggest, at the very least, that this important limitation be stated.

Reply: We thank the reviewer for this important comment. Indeed, risk results from the combination of hazard, exposure and vulnerability, and in our analysis we do not take into account vulnerability. In the following we explain our reasoning why the analysis remains valid and how we revised the text in order to address this comment.

Adaptation is a key factor in reducing vulnerability to climate physical risks. However, data on firms' adaptation strategies are very scarce or not available at all. In this regard, a comprehensive analysis of reports to the Carbon Disclosure Project (CDP) by (Goldstein et al., 2019) showed that "companies report the costs of both physical climate change impacts and the strategies required to manage them sporadically and inconsistently, while the strategies themselves overall reflect a narrow view of risk that underestimates supply chain and broader societal impacts." Thus, with information about vulnerability and adaptation missing at the firm-level, we cannot dispose of asset-level vulnerability and adaptation data.

We acknowledge that methods to appraise adaptation options exist, for example in CLIMADA (Bresch and Aznar-Siguan, 2021) or in (Mandel et al., 2021). However, these methods are underpinned by assumptions that are very difficult to calibrate. For example, to properly leverage the adaptation appraisal methods in CLIMADA (Bresch and Aznar-Siguan, 2021) one would need to set multiple parameters speaking to the type of adaptation measures, their costs, and their effect in mitigating hazards. Calibrating such parameters is complex, in lack of public and private sector information. For example, in (Rana et al., 2022), only one adaptation measure is assumed to be impacting tropical cyclones, namely mangroves. The authors assume that mangroves plantation or restoration would decrease the wind intensity for tropical cyclones by 4m/s, but there is no information on how this assumption is calibrated. Without this information, it is difficult to properly account for adaptation options.

Moreover, adaptation efforts are extremely limited in Mexico, both from private firms and the public sector. In fact, (Escudero and Mendoza, 2021) concludes that “Unfortunately, although some successful efforts exist, Mexico lacks significant adaptation actions, while several phenomena related to climate change affect and degrade its coastal system.” At the same time, Climate Action Tracker (<https://climateactiontracker.org/countries/mexico/>) notes that for Mexico “the government allocated most of the ‘climate change mitigation and adaptation’ budget in 2021 and 2022 to fossil gas transport infrastructure”. As a consequence, the lack of adaptation investment and calibration data would make the consideration of adaptation in our analysis an arbitrary exercise.

Nevertheless, we acknowledge that in (Bresch and Aznar-Siguan, 2021, Mandel et al., 2021, Rana et al., 2022) the introduction of adaptation measures significantly reduces the damages borne by sovereigns. Thus, we can expect that adaptation would reduce losses also in our results, at the portfolio level.

To address the reviewer’s comment we have now acknowledged explicitly this limitation in the text and its expected effects on the results, in “Methods”, subsection “Limitations”, as follows:

“Third, we do not consider firms’ adaptation measures (such as sea barriers or mangroves) due to lack of data on firms’ adaptation strategies (Goldstein et al., 2019). Adaptation may vary across firms and different firms may follow different schedules to implement adaptation measures. Moreover, many adaptation measures generally considered in the literature (e.g. mangroves for coastal protection, as in (Rana et al., 2022)) are not relevant for the types of assets that we analyze here. Similarly, relocation is not feasible for most of the assets we consider (e.g., mines or power plants must be located where natural resources are located). Existing calibrations of adaptation measures are either based on assumptions (e.g., Rana et al., 2022 in the case of the effect of mangroves on tropical cyclones’ winds) or specific to individual countries and thus not applicable to Mexico (e.g. Bresch and Aznar-Siguan, 2021). Furthermore, Mexico invests very little in adaptation (Climate Action Tracker, 2022, Escudero and Mendoza, 2021).”

The probabilistic assessment of damages from tropical cyclones

The authors undertake a probabilistic assessment of damages from TCs based on Aznar-Siguan & Bresch (2019), which includes the introduction of synthetic cyclone tracks to

historical cyclone tracks to overcome the problem of such events occurring only infrequently. These tracks are then perturbed in CLIMADA, as described in Knutson et al (2015), through the dynamical downscaling of global climate projections across RCPs 2.5, 4.5 and 6.0 for the years 2035, 2040 and 2050.

There are for me several issues that I will address each in turn.

R3C2

- Timescale. In contrast to the work of Knutson et al (2015), who run simulations for present day (2001-20) and late twenty-first century (2081-2100), the authors attempt to assess changes in TC frequency and intensity across the years 2035, 2040 and 2050. As noted in Knutson et al (2015), however, changes in TC frequency and intensity at decadal scales cannot be determined because natural variability (i.e., influences on weather events from phenomena like the El Nino Southern Oscillation) is not well understood, even if the longer-term outputs of the average changes of these is. For this reason, most climate projections across different RCPs tend not to be distinguishable until around 2050 onwards. This raises questions for the results illustrated in Tables I II for 2040.

Reply: We thank the reviewer for the opportunity to clarify the role of time in our model. There may be a misunderstanding here, as we *do not* “assess changes in tropical cyclones’ (TC) frequency and intensity across the years 2035, 2040, 2045”, nor our results depend on the differences between assessments of the impact of TC across those years. Our model uses only one year for estimates, i.e. 2040, while other years provide robustness checks. So our results are not affected by the fact that TC impacts in 2035, 2040, 2045 cannot be distinguished with statistical significance.

More in detail, the model we use to compute the value of the firm (an adaptation of a standard dividend discount model) assumes that only the long-term growth rate of the firm from 2035 onward is affected by climate impacts (i.e. it is lower than in the absence of them). The impacts from physical risks, and in particular from tropical cyclones, are proxied by the values computed by the CLIMADA and ICES models for a selected reference year, in this case 2040, which enters the calculations that yield the results presented in Table II. This approach is consistent with the literature, which most often uses a reference year (see for example (Bresch and Aznar-Siguan, 2021) and (Rana et al., 2022), both using 2050 as reference year). We also tested the computations using as references the years 2035, 2045, and 2050, and we did not find significant differences (see results reported in Table A below). As such, the variability between 2035, 2040, 2045, and 2050 does not play a role in our model, and the fact that the impacts from tropical cyclones are indistinguishable among these years does not invalidate our results.

We believe that this approach is consistent with the current climate risk assessment literature (e.g. Bresch and Aznar-Siguan, 2021, Rana et al., 2022). In fact, while tropical cyclone intra-decadal variability cannot be precisely assessed, we know their frequency and intensity will increase by the end of the century (Knutson et al., 2015). Without further information on the changes around 2050, a linear interpolation approach such as the one implemented in CLIMADA represents a reasonable approximation and was thus chosen for our study. In the linear interpolation approach, asset-level damages increase between

historical and 2035-2050 for scenarios Representative Concentration Pathways (RCP) RCP4.5 and 6.0, and decrease between 2035 and 2050 for scenario RCP2.6, while still being higher than historical damages. As such, we believe the approach to use a reference year for future tropical cyclones risk is consistent with the literature and justifiable.

To address this comment of the reviewer and avoid any confusion for the readers, we have now clarified the role of time in the “Methods”, with the addition of the following paragraph to the subsection “Climate Dividend Discount Model” (subsection “Climate financial valuation: the Climate Dividend Discount Model”):

“In our model, the computation of the value of the firm takes into account the year span from 2022 to 2050. In the long run (i.e. from 2035 onward) we assume firms are subject to climate impacts. To proxy these impacts, we use a reference year for both the ICES model and tropical cyclones, namely 2040 (climate models’ estimates at years 2035, 2040, 2045, 2050 are not distinguishable anyway in statistical sense, (Knutson et al., 2015)). In particular, the estimate of TC impacts at 2040 is obtained following a common approach in the literature based on a linear approach interpolation of the impacts between 2020 and 2100 (see e.g. (Bresch and Aznar-Siguan, 2021, Rana et al., 2022)).

Thus, for the valuation conditioned to e.g. scenario SSP2-RCP6.0, year 2040, ICES data are considered for SSP2-RCP6.0, year 2040, and tropical cyclones impacts are considered for RCP6.0, year 2040. Other years are not considered.”

Scenario	EAI (%)	RP250 (%)
Overall	-0.75	-3.7
SSP2-RCP6.0, 2035	-0.98	-3.6
SSP2-RCP6.0, 2040	-0.95	-3.7
SSP2-RCP6.0, 2045	-0.53	-3.6
SSP2-RCP6.0, 2050	-0.17	-3.4
SSP3-RCP2.6, 2035	-1.1	-3.7
SSP3-RCP2.6, 2040	-0.92	-3.6
SSP3-RCP2.6, 2045	-0.73	-3.4
SSP3-RCP2.6, 2050	-0.52	-3.2
SSP3-RCP4.5, 2035	-0.94	-3.8
SSP3-RCP4.5, 2040	-0.84	-3.9
SSP3-RCP4.5, 2045	-0.70	-3.9
SSP3-RCP4.5, 2050	-0.53	-4.0

SSP5-RCP4.5, 2035	-0.95	-3.8
SSP5-RCP4.5, 2040	-0.82	-3.9
SSP5-RCP4.5, 2045	-0.72	-4.0
SSP5-RCP4.5, 2050	-0.59	-4.0

Table A: Portfolio-level results, equally weighted across scenarios (“Overall”) and for each scenario and year combination. “Chronic + RP250, asset-level” case only. First column: scenario specifications. Second column: portfolio loss, EAI (mean). Third column: portfolio loss, RP250 (VaR). Source: authors.

R3C3

- **Weather.** As also noted in Knutson et al (2015), as well as by CLIMADA (2017), the influence of local features and conditions such as cloud cover, topography, sea-surface temperature and humidity mean extreme events such as TCs develop at a scale not well represented in global climate model (GCM) projections, which CLIMADA data are based on. Accordingly, “how hurricane damage changes with climate remains challenging to assess” (CLIMADA, 2017). This similarly raises questions for how TCs will actually develop for the year 2040.

Reply: We thank the reviewer for the comment. In a similar spirit to the reply to the previous comment, we would like to emphasize that our results do not depend on the accuracy of the *changes* in hurricane damages over the next two decades. Our results are driven by how the value of the firm varies when 1) neglecting future climate impacts, 2) considering chronic risk only, and 3) considering both chronic and acute risk (only hurricanes, though).

We know that frequencies and intensities of tropical cyclones will change, heterogeneously across basins, by the end of the century (Knutson et al., 2015). For the estimation of the impact of hurricanes in our framework we use only a middle of the century estimate, namely 2040. To compute this estimate, we use the linear interpolation method implemented in CLIMADA, which is widely applied in the literature (Aznar-Siguan and Bresch, 2019, Bresch and Aznar-Siguan, 2021, Rana et al., 2022). This enables us to obtain financially-relevant outcomes from the probabilistic risk assessment of damages.

We have now further elaborated on this point in the “Discussion” section:

“Third, in order to quantify tropical cyclones’ impacts, we relied on traditional approaches in the literature (e.g. (Bresch et al., 2021, Rana et al., 2022)) that interpolate between current (2001-2020) and future (2081-2100) climate scenarios but do not take into account the uncertainty on the evolution of the impacts.”

and “Methods”, subsection “Limitations”:

“Second, the reader should be aware that there is considerable uncertainty regarding the effects of climate change on the frequency and intensity of tropical cyclones for the middle of the century especially at less-than continental scale. The methodology we applied here to

quantify damages around the middle of the century builds on relevant literature in the field and relies on interpolation (e.g., (Bresch et al., 2021, Rana et al., 2022)).“

(see also the replies to subsequent comments).

R3C4

- **Uncertainty.** Any probabilistic assessment of a rare event based on global climate model simulations must include uncertainty bounds. My reading of the paper suggests confidence intervals at the 95th percentile are provided for the bootstrapping of average portfolio loss and VaR. However, it is not clear to me that uncertainties relating to the underlying GCMs and the likelihood of actual TC impacts (as discussed in the prior two points) propagates through into the estimation of investor losses illustrated in Tables I and II.

Reply: We thank the reviewer for the very relevant comment. Indeed, our analysis only accounts for the sources of uncertainty that we can quantify at this stage. The results are to be interpreted conditional to this simplification. We have now specified in our “Discussion” which sources of uncertainty are taken into account and which ones are not, with the following sentences:

“Our results are conditioned to the following limitations, some of which, however, apply more in general to the field of climate physical risk assessment.

We acknowledge that our analysis does not take into account all the sources of uncertainty. Thus our results provide a quantification of losses conditional to the following specifications.

First, we rely on point-estimates of the characteristics of assets (e.g., residual life, monetary value, capacity) because confidence intervals on those estimates are not available.

Second, we take the point-estimate for Return Period (RP) and Expected Annual Impacts (EAI) of tropical cyclones (TC) impacts from the CLIMADA model under the standard setting (Aznar-Siguan and Bresch, 2019). Indeed, there is not yet an established way to calibrate the set-up that could yield uncertainty on the estimates of RP and EAI (Kropf et al., 2022). In addition, the data needed to properly calibrate assumptions about uncertainty is not available.

Third, to quantify the impact of tropical cyclones, we relied on traditional approaches in the literature (e.g. Bresch and Aznar-Siguan, 2021, Rana et al., 2022) that interpolate between current (2001-2020) and future (2081-2100) climate scenarios but they do not take into account the uncertainty on the evolution of the impacts.

Furthermore, we use the same damage function across all types of assets since data on damages by asset-type are not available. In this regard, publicly available datasets, such as EM-DAT (UCLouvain, 2009) only provide damages per event, at the country or (more rarely) at the subnational level.

Finally, we do not consider firms' adaptation efforts (such as assets' relocation, implementation of physical barriers, etc.) because this information is currently not available (Goldstein et al., 2019).

Further limitations are discussed in Methods.”

Advice with respect to publication

R3C5

In conclusion, the results of this paper are without a doubt of interest and possibly indicative. My suggestion for publication would be for the authors to amplify the limitations and uncertainties arising from the confounding variables both the climate and the tools of climate science introduce, however.

Reply: We thank the reviewer for the encouraging comment. As mentioned in the reply to the previous comment, we have now clarified in the “Discussion” the limitations of our work in particular in terms of the quantification of uncertainty. Further, in “Methods” (subsection “Limitations”) we have provided complimentary remarks on the limitations of the results elaborated on their implications for climate physical risk assessment.

R3C6

Specifically, a significant contribution could be made to the literature, but also to policy-making more broadly, if the opportunity were taken in this paper to discuss the limitations of using climate science in its current form for the purpose of simulating the physical effects of extreme events at less-than continental-scales and for the years 2030-2050 in particular. Considerable advances and investment in climate modelling, as well as in co-operation between the climate modelling and economic communities, are needed if we are to develop the capacity to understand the financial effects of climate change for portfolios at the scale of a single asset more reliably. There is significant research potential in such collaborative work for both economics and science, and this article in this journal could have the effect of motivating such work.

Reply: We thank the reviewer for the encouragement to take the opportunity to raise the attention on the importance of the need for better cooperation between the climate modelling and economic communities. We have now acknowledged in “Discussion” that there are uncertainties related to climate modelling that are not considered in our analysis (see previous reply).

Further remarks in “Methods” (subsection “Limitations”) have been rewritten to address the reviewer’s comment:

“The following remarks complement the limitations acknowledged in the “Discussion”.

First, we account for uncertainties on financial portfolio loss (VaR, mean) estimating the confidence intervals (CI) using bootstrapping (see e.g. (DiCiccio and Efron, 1996)). Including the sources of uncertainties mentioned in the “Discussion” would likely lead to larger CI.

Second, the reader should be aware that there is considerable uncertainty regarding the effects of climate change on the frequency and intensity of tropical cyclones for the middle of the century especially at less-than continental scale. The methodology we applied here to quantify damages around the middle of the century builds on relevant literature in the field and relies on interpolation (e.g. Bresch and Aznar-Siguan, 2021, Rana et al., 2022).

Third, we do not consider firms' adaptation measures (such as sea barriers or mangroves) due to lack of data on firms' adaptation strategies (Goldstein et al., 2019). Adaptation may vary across firms and different firms may follow different schedules to implement adaptation measures. Moreover, many adaptation measures generally considered in the literature (e.g. mangroves for coastal protection, as in (Rana et al., 2022)) are not relevant for the types of assets that we analyze here. Similarly, relocation is not feasible for most of the assets we consider (e.g., mines or power plants must be located where natural resources are located). Existing calibrations of adaptation measures are either based on assumptions (e.g., Rana et al., 2022 in the case of the effect of mangroves on tropical cyclones' winds) or specific to individual countries and thus not applicable to Mexico (e.g. Bresch and Aznar-Siguan, 2021). Furthermore, Mexico invests very little in adaptation (Climate Action Tracker, 2022, Escudero and Mendoza, 2021).

Fourth, information on assets' location, ownership, value and residual life is often missing and has to be estimated. Moreover, some non-core firms' assets (e.g., deposits, warehouses) may be unknown even for firms where asset-level data are available. Furthermore, for some assets it is not possible to reconstruct ownership chains, or the unlisted nature of some of the owners makes the link to equity financial portfolios not possible.

Fifth, in our assessment we consider only one country (Mexico), one hazard (tropical cyclones), a selection of asset types (mostly energy-related), and one financial asset class (equities). Thus, our results in terms of financial risk for investors are conservative.

Sixth, short-term risks are generally downplayed both in the macroeconomic model framework used (see (Eboli et al., 2010) for a discussion), and in the CDDM.

Finally, we consider only direct impacts of tropical cyclones, and not their indirect ones such as supply chain disruptions, damage to infrastructure other than the assets in the sample, or loss of lives.”

Accounting for these sources of uncertainty would lead in principle to larger confidence intervals on the final estimates for portfolio-level losses and for the Value at Risk (VaR). However, data to calibrate the distributions underpinning the uncertainty analysis (e.g. reasonable bounds of the uncertainty around the threshold of a damage function) are currently missing, and incorrect calibrations would give a false sense of security. Thus, we opted for recognizing the limitations rather than providing a quantification based on further assumptions.

Finally, we have added a paragraph in the “Discussion” to highlight the need for closer collaboration between climate scientists and economic/finance experts in the context of physical risk assessment, as follows:

“Finally, further investments in climate physical risk models are necessary to meet the needs of the financial sector (Fiedler et al., 2021). At the same time, closer collaboration between climate modelers and economists is necessary to improve climate physical risk assessment at the asset, firm, and portfolio levels, ultimately enabling better investment and policy decisions. However, the quest for better models is no justification to delay action by investors and financial supervisors in assessing physical risks and climate change adaptation on the basis of available models.”

References cited

Aznar-Siguan, G., & Bresch, D. N. (2019). CLIMADA v1: a global weather and climate risk assessment platform. *Geoscientific Model Development*, 12(7), 3085–3097.

<https://doi.org/10.5194/gmd-12-3085-2019>

Cardona, O.-D., van Aalst, M. K., Birkmann, J., Fordham, M., McGregor, G., Perez, R., Pulwarty, R. S., Schipper, E. L. F., Sinh, B. T., Décamps, H., Keim, M., Davis, I., Ebi, K. L., Lavell, A., Mechler, R., Murray, V., Pelling, M., Pohl, J., Smith, A.-O., & Thomalla, F. (2012). Determinants of Risk: Exposure and Vulnerability. In *Managing the Risks of Extreme Events and Disasters to Advance Climate Change Adaptation* (pp. 65–108). Cambridge University Press. <https://doi.org/10.1017/CBO9781139177245.005>

CLIMADA. (2017). Hazard: Tropical cyclones.

https://climada-python.readthedocs.io/en/stable/tutorial/climada_hazard_TropCyclone.html

Knutson, T. R., Sirutis, J. J., Zhao, M., Tuleya, R. E., Bender, M., Vecchi, G. A., Villarini, G., & Chavas, D. (2015). Global Projections of Intense Tropical Cyclone Activity for the Late Twenty-First Century from Dynamical Downscaling of CMIP5/RCP4.5 Scenarios. *Journal of Climate*, 28(18), 7203–7224. <https://doi.org/10.1175/JCLI-D-15-0129.1>

References cited in the replies

- (Aznar-Siguan and Bresch, 2019) Aznar-Siguan, G., & Bresch, D. N. (2019). *CLIMADA v1: a global weather and climate risk assessment platform*. *Geoscientific Model Development*, 12 (7), 3085–3097.
- (Bresch and Aznar-Siguan, 2021) Bresch, D. N., & Aznar-Siguan, G. (2021). *Climada v1. 4.1: Towards a globally consistent adaptation options appraisal tool*. *Geoscientific Model Development*, 14 (1), 351–363.
- (Climate Action Tracker, 2022) Climate Action Tracker. (2022). *Country summary: Mexico*.
- (DiCiccio and Efron, 1996) DiCiccio, T. J., & Efron, B. (1996). *Bootstrap confidence intervals*. *Statistical Science*, 11 (3), 189–228.
- (Eboli et al., 2010) Eboli, F., Parrado, R., & Roson, R. (2010). *Climate-change feedback on economic growth: explorations with a dynamic general equilibrium model*. *Environment and Development Economics*, 15 (5), 515–533.
- (Escudero and Mendoza, 2021) Escudero, M., & Mendoza, E. (2021). *Community perception and adaptation to climate change in coastal areas of Mexico*. *Water*, 13 (18), 2483.
- (Fiedler et al., 2021) Fiedler, T., Pitman, A. J., Mackenzie, K., Wood, N., Jakob, C., & Perkins-Kirkpatrick, S. E. (2021). *Business risk and the emergence of climate analytics*. *Nature Climate Change*, 11 (2), 87–94.

- (Goldstein et al., 2019) Goldstein, A., Turner, W. R., Gladstone, J., & Hole, D. G. (2019). *The private sector's climate change risk and adaptation blind spots*. *Nature Climate Change*, 9 (1), 18–25.
- (Knutson et al., 2015) Knutson, T. R., Sirutis, J. J., Zhao, M., Tuleya, R. E., Bender, M., Vecchi, G. A., Villarini, G., & Chavas, D. (2015). *Global projections of intense tropical cyclone activity for the late twenty-first century from dynamical downscaling of cmip5/rcp4.5 scenarios*. *Journal of Climate*, 28 (18), 7203–7224.
- (Kropf et al., 2022) Kropf, C. M., Ciullo, A., Otth, L., Meiler, S., Rana, A., Schmid, E., McCaughey, J. W., & Bresch, D. N. (2022). *Uncertainty and sensitivity analysis for probabilistic weather and climate-risk modelling: an implementation in CLIMADA v.3.1.0*. *Geoscientific Model Development*, 15 (18), 7177–7201.
- (Mandel et al., 2021) Mandel, A., Tiggeloven, T., Lincke, D., Koks, E., Ward, P., & Hinkel, J. (2021). *Risks on global financial stability induced by climate change: the case of flood risks*. *Climatic Change*, 166(1-2), 4.
- (Rana et al., 2022) Rana, A., Zhu, Q., Detken, A., Whalley, K., & Castet, C. (2022). *Strengthening climate resilient development and transformation in Viet Nam*. *Climatic Change*, 170 (1-2), 4.

REVIEWER COMMENTS

Reviewer #1 (Remarks to the Author):

I did already comment on the paper in round one and find all concerns being properly addressed.

Reviewer #4 (Remarks to the Author):

Review for Asset-level assessment of climate physical risk matters for adaptation finance
Nature Communications

This paper quantifies financial and economic losses stemming the physical climate risk exposure of firms' real assets. Just like other referees who have reviewed this manuscript prior to me, I also feel that this is more than worthwhile to do because the accurate estimation of exposure to financial losses through physical risks is understudied in the literature. My sense always has been that this is due to the difficulty of identifying real assets and connecting these real assets to firms in the cross-section and over time. Since I am replacing R3 in this round, my only comment will be qualitative in nature in this direction and it is ultimately up to the authors to implement it or not.

- I think the main takeaway from the paper is summarized perfectly in pg. 10: "Neglecting asset-level information leads to a relevant underestimation of losses for investors." While this is true, it is important to highlight that institutional investors tend to know much more about their portfolio firms. The location of real assets of firms is often unobservable to the researcher but it doesn't necessarily follow that this is unobservable to investors as well. In fact, there is some evidence in the literature that investors (or even households) care about these long-run physical risk exposures when making financial decisions, indicating that this information is available to these financial agents. As such, I think it would be a good idea to underline the fact that this improvement may be most relevant for researchers and policymakers rather than investors.

- Second, given that the improvement in identifying the location of firms' real assets takes

such a central place in the paper, I encourage the authors to more prominently share their procedure, data sources, and methodology on how they formed this database. As they also admit, the accuracy of estimation of future financial losses crucially depends on the data on location of real assets and this is no different for this manuscript either. I read that their database returns 123,340 physical assets globally and only 3,493 of these are located in Mexico. I am not sure why the authors chose to focus on Mexico or European investors, but providing extensive information on guidance on how to form such databases would make the paper more impactful in my opinion.

-- Some explanation on why Mexico was chosen as the focus of the study would also be great to see, especially since the authors also seem to have such granular data presumably on other suitable countries with a lot of exposure to physical climate risks.

Comments of R3 and Responses

I believe R3 lists a number of concerns regarding the limitations of the methodology employed in the paper. R3C1 brings up the point that the authors do not consider the heterogeneity in vulnerability of different assets to the physical risks considered in the paper. The authors argue that there is insufficient data regarding adaptation on the firm side and adaptation efforts are extremely limited in Mexico. My sense was that the authors mostly considered man-made adaptation measures in this argument and I would think that natural "adaptation" mechanisms could also be important, especially we since Mexico is a country with a large level of variation when it comes to topography compared to, say, the Netherlands. As the authors suggest in their response, such adaptation and mitigation strategies, be it man-made or natural, would reduce the estimated financial losses. I think acknowledging the limits of the method and data is a good first step but I think the manuscript would also benefit from a back of the envelope kind of calculation on the errors the financial loss estimation may inherently have due to these missing data.

Similarly, R3C4 raises a concern about the uncertainty bounds associated with the global climate model (GCM) simulations. The authors acknowledge the limitations of their methodology and state that they are only able to take into account the sources of uncertainty they can quantify. Thus, we can think of the estimates in the paper as having

some “error bounds” around them associated with such limitations. The authors do not elaborate on how large these error bounds may be nor do they discuss how the size of such error bounds may be sensitive to their different assumptions. Another way of saying is that the reader does not get a sense of how sensitive the final estimates are to different assumptions made in the paper. This is not just true for R3C4 but for R3C1 through R3C4. Even if only crude understanding of the sensitivity of the results to these assumptions can be provided, I think the authors are better off providing such discussion to give the reader a sense of robustness and direction.

REVIEWERS' COMMENTS

Reviewer #4 (Remarks to the Author):

I thank the authors for carefully responding to all comments and I think my comments were sufficiently answered.